# A targetable CoQ-FSP1 axis drives ferroptosis- and radiation-resistance in KEAP1 inactive lung cancers

Pranavi Koppula [1,2,6], Guang Lei [1,6], Yilei Zhang [1], Yuelong Yan[1], Chao Mao[1], Lavanya Kondiparthi[3], Jiejun Shi [4], Xiaoguang Liu [1], Amber Horbath [1], Molina Das[1], Wei Li [4], Masha V. Poyurovsky[3], Kellen Olszewski[3,5] & Boyi Gan [1,2✉]

Targeting ferroptosis, a unique cell death modality triggered by unrestricted lipid peroxidation, in cancer therapy is hindered by our incomplete understanding of ferroptosis mechanisms under specific cancer genetic contexts. *KEAP1* (kelch-like ECH associated protein 1) is frequently mutated or inactivated in lung cancers, and *KEAP1* mutant lung cancers are refractory to most therapies, including radiotherapy. In this study, we identify ferroptosis suppressor protein 1 (FSP1, also known as AIFM2) as a transcriptional target of nuclear factor erythroid 2-related factor 2 (NRF2) and reveal that the ubiquinone (CoQ)-FSP1 axis mediates ferroptosis- and radiation- resistance in *KEAP1* deficient lung cancer cells. We further show that pharmacological inhibition of the CoQ-FSP1 axis sensitizes *KEAP1* deficient lung cancer cells or patient-derived xenograft tumors to radiation through inducing ferroptosis. Together, our study identifies CoQ-FSP1 as a key downstream effector of KEAP1-NRF2 pathway and as a potential therapeutic target for treating *KEAP1* mutant lung cancers.

[1] Department of Experimental Radiation Oncology, The University of Texas MD Anderson Cancer Center, Houston, TX 77030, USA. [2] The University of Texas MD Anderson UTHealth Graduate School of Biomedical Sciences, Houston, TX 77030, USA. [3] Kadmon Corporation, LLC, New York, NY 10016, USA. [4] Division of Computational Biomedicine, Department of Biological Chemistry, School of Medicine, University of California, Irvine, CA 92697, USA. [5] Present address: The Barer Institute, Philadelphia, PA 19104, USA. [6] These authors contributed equally: Pranavi Koppula, Guang Lei. ✉email: bgan@mdanderson.org

Ferroptosis is a form of iron-dependent lipid peroxidation-induced cell death that differs from other forms of regulated cell death, such as apoptosis, due to its unique mechanistic basis and morphological features[1–5]. Mechanistically, eukaryotic cells are under constant assault by lipid peroxides that are generated by peroxidation of polyunsaturated-fatty-acid-containing phospholipids (PUFA-PLs) in cellular membranes under iron- and reactive oxygen species (ROS)-rich conditions; if left unrepaired, these toxic lipid peroxides can accumulate to lethal levels, damage the integrity of cellular membranes, and ultimately trigger ferroptotic cell death[4]. Morphologically, ferroptotic cells do not exhibit typical features shown in other forms of regulated cell death, such as chromatin condensation, apoptotic body or autophagosome formation, but contain mitochondria with increased membrane density and diminished cristae[3,4].

Cells have evolved diverse ferroptosis surveillance mechanisms to defend against ferroptosis and to maintain cell viability. One major ferroptosis surveillance system is mediated by the solute carrier family 7 member 11-reduced glutathione-glutathione peroxidase 4 (SLC7A11-GSH-GPX4) signaling axis, wherein extracellular cystine is taken up by SLC7A11 for GSH biosynthesis, and GSH is subsequently utilized by GPX4 to detoxify lipid hydroperoxides to lipid alcohol (Fig. 1a)[3,6–8]. In addition, ferroptosis suppressor protein 1 (FSP1; also called AIFM2 or AMID) and dihydroorotate dehydrogenase (DHODH) reduce ubiquinone (CoQ) to ubiquinol (CoQH$_2$) on the plasma membrane and inner mitochondrial membrane, respectively; CoQH$_2$ can neutralize lipid peroxyl radicals as a radical-trapping antioxidant, resulting in ferroptosis suppression (Fig. 1a)[9–11]. Inactivation of these ferroptosis surveillance systems by ferroptosis inducer (FIN) compounds leads to a rapid buildup of lipid peroxides and triggers potent ferroptosis in many cancer cell lines and in vivo tumor models[1,4].

Accumulating evidence indicates that ferroptosis is a critical tumor suppression mechanism[12,13]. Ferroptosis at least partly mediates tumor-suppressive activities of several tumor suppressors such as p53 and BAP1[14,15]. Ferroptosis inactivation has been shown to contribute to oncogenic Kras-driven tumor progression[16] and tumor metastasis[17]. This has spurred much interest in cancer research communities to further target ferroptosis in cancer therapy[12,13]. Studies in recent years have explored FINs to treat cancers that are particularly susceptible to ferroptosis[18–20] or proposed to combine FINs with other standards of care, such as radiotherapy (RT), in cancer treatment[21–25]. However, due to enormous genetic heterogeneities in cancer, targeting ferroptosis in any given cancer will require a deeper understanding of ferroptosis regulatory mechanisms in corresponding cancer genetic contexts.

Tumor suppressor kelch-like ECH associated protein 1 (KEAP1) is a substrate adapter in the KEAP1-Cullin-3 ubiquitin ligase complex, which targets nuclear factor erythroid 2-related factor 2 (NRF2) for proteasomal degradation under unstressed conditions. Oxidative stress or KEAP1 inactivation (by its loss-of-function mutation, deletion, or epigenetic silencing in cancers) stabilizes NRF2, which subsequently translocates into the nucleus and promotes the transcription of a host of genes governing antioxidant defense and redox maintenance[26–28], including genes involved in ferroptosis suppression[29]. KEAP1 is mutated in around 16% of non-small cell lung cancers (NSCLCs), including 12% lung squamous cell carcinomas (LUSCs)[30] and 20% lung adenocarcinomas (LUADs)[31]. Lung cancer patients with KEAP1 mutations have shorter overall survival with poor prognosis, and KEAP1 mutant lung tumors are refractory to most available standard-of-care therapies including RT[32–36], highlighting the pressing need to develop novel and effective combination therapies for this type of lung cancer.

In this work, we identify FSP1 as an NRF2 transcriptional target and reveal that KEAP1 mutation or deficiency in lung cancer cells upregulates FSP1 expression through NRF2, leading to ferroptosis- and radiation- resistance. We further show that targeting the CoQ-FSP1 axis sensitizes KEAP1 mutant lung cancer cells or tumors to radiation by inducing ferroptosis, thereby identifying a novel therapeutic strategy to target ferroptosis in KEAP1 mutant lung cancers.

## Results

**KEAP1 deficiency in lung cancer cells has differential effects on ferroptosis induced by different FINs.** There are at least three classes of FINs: class 1 FINs (such as erastin) and class 2 FINs (such as RSL3 and ML162) induce ferroptosis by inhibiting SLC7A11 and GPX4, respectively, whereas class 3 FIN (FIN56) acts by depleting both GPX4 protein and CoQ (Fig. 1a)[3,7,37]. To understand the role and mechanisms of KEAP1 in regulating ferroptosis, we examined the effect of KEAP1 deficiency on ferroptosis sensitivity to different classes of FINs in lung cancer cells. We confirmed that KEAP1 deletion markedly increased levels of NRF2 and its transcriptional target SLC7A11[38] in H1299 cells (a KEAP1 wild-type [WT] lung cancer cell line) (Fig. 1b and Supplementary Fig. 1a), and correspondingly rendered H1299 cells resistant to ferroptosis induced by erastin (Fig. 1c). Surprisingly, KEAP1 deletion in H1299 cells dramatically decreased GPX4 levels (Fig. 1b) in an NRF2-dependent manner, as deleting NRF2 in KEAP1 deficient H1299 cells restored GPX4 expression (Supplementary Fig. 1b). Notably, despite the decreased GPX4 expression in KEAP1 deficient H1299 cells, KEAP1 deletion significantly promoted ferroptosis resistance to RSL3 or ML162 (Fig. 1d and Supplementary Fig. 1c). We further confirmed that erastin- or RSL3-induced cell death in H1299 cells was abolished by the ferroptosis inhibitor ferrostatin-1, but not by the apoptosis inhibitor Z-VAD-FMK or the necroptosis inhibitor necrostatin-1s (Supplementary Fig. 1d, e). In contrast, KEAP1 deletion in H1299 cells did not affect ferroptosis sensitivity to FIN56 (Fig. 1e, f). Lipid peroxidation is a hallmark of ferroptosis[4]. Consistently, KEAP1 deficiency attenuated lipid peroxidation induced by erastin, RSL3, or ML162, but not by FIN56 (Fig. 1g, h and Supplementary Fig. 1f–k).

Ostensibly, decreased GPX4 levels in KEAP1 deficient H1299 cells would not explain why these cells are resistant to RSL3 (which inhibits GPX4); on the other hand, increased SLC7A11 levels in KEAP1 deficient cells could be responsible for the increased resistance of these cells to RSL3 (Fig. 1b, d). However, we found that knocking down SLC7A11 in KEAP1 knockout (KO) H1299 cells did not significantly affect ferroptosis sensitivity to RSL3 in KEAP1 KO cells, although SLC7A11 knockdown in H1299 cells did significantly promote RSL3-induced ferroptosis (Fig. 1i, j). We made similar observations by using ML162 (Fig. 1k) and further confirmed this observation by lipid peroxidation measurement (Fig. 1l, m). Further, while deleting GPX4 in H1299 cells induced massive lipid peroxidation and ferroptosis (which could be fully suppressed by the ferroptosis inhibitor ferrostatin-), KEAP1 KO counterparts with GPX4 deficiency did not exhibit obvious lipid peroxidation or cell death and could be cultured without ferrostatin-1 (Fig. 1n–q). These data suggest that, in H1299 cells, KEAP1 deficiency promotes resistance to ferroptosis induced by class 2 FINs independent of SLC7A11 or GPX4.

We made similar observations in another KEAP1 WT lung cancer cell line H23 cells: KEAP1 deletion rendered H23 cells resistant to ferroptosis induced by erastin, RSL3, but not FIN56 (Supplementary Fig. 1l–o); SLC7A11 knockdown promoted RSL3- or ML162-induced ferroptosis in H23 cells but not in

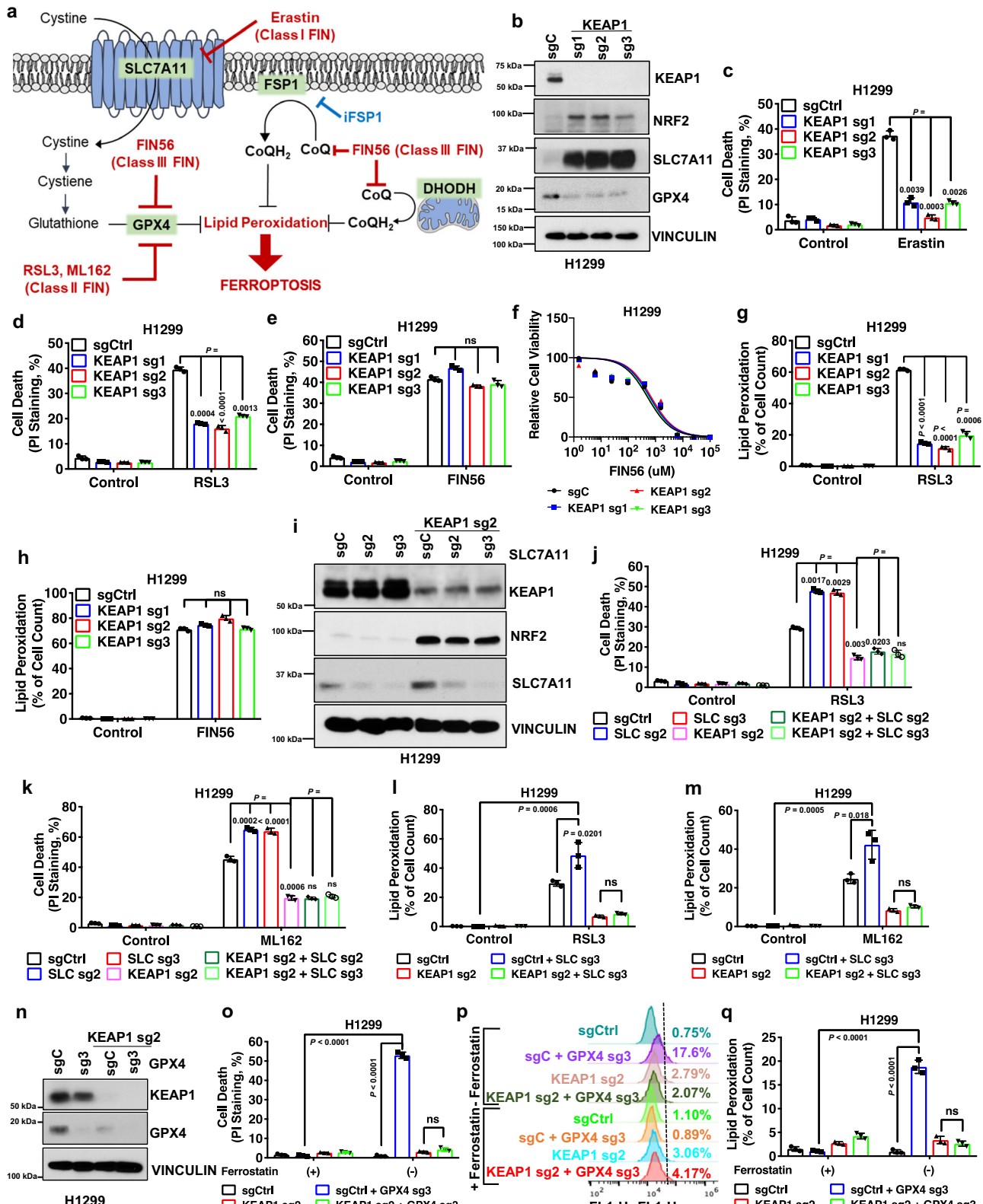

KEAP1 KO counterparts (Supplementary Fig. 1p–r). It should be noted that *KEAP1* deletion increased GPX4 levels in H23 cells (Supplementary Fig. 1l). Unlike *SLC7A11*, *GPX4* is not considered a classical NRF2 transcriptional target in the literature[39–41]. Based on our data and a recent study[42], it appears that *KEAP1* deficiency can either suppress or promote GPX4 expression in a context-dependent manner. In either case, *KEAP1* deletion

promotes ferroptosis resistance to class 2 FINs (RSL3 and ML162), but not to the class 3 FIN (FIN56).

Conversely, WT *KEAP1* restoration in *KEAP1* mutant H460 cells promoted ferroptosis induced by RSL3 and ML162 (Supplementary Fig. 2a–c) but not to FIN56 (Supplementary Fig. 2d). Likewise, *KEAP1* restoration in H460 cells promoted lipid peroxidation induced by RSL3 or ML162 (Supplementary

**Fig. 1 KEAP1 regulates ferroptosis in a SLC7A11/GPX4-independent manner in lung cancer cells. a** Schematic overview of ferroptosis pathways and ferroptosis inducers (FINs) was used in this study. Ferroptosis is induced by lipid peroxide buildup. SLC7A11 imports extracellular cystine. Intracellular cystine then undergoes reduction to generate cysteine, which is the rate-limiting precursor required for glutathione (GSH) synthesis. GSH serves as a cofactor for GPX4 to detoxify lipid peroxides. Ubiquinol ($CoQH_2$) can detoxify lipid peroxyl radicals and can be generated from ubiquinone (CoQ) by FSP1 and DHODH on the plasma membrane and inner mitochondrial membrane, respectively. FINs are classified into various classes depending on their targets of action. Class I FIN used in this study, erastin, targets SLC7A11-mediated cystine import. Class II FINs used in this study, RSL3, and ML162, target GPX4 activity. Class III FIN used in this study, FIN56, depletes both GPX4 protein, and $CoQH_2$. **b** Protein levels of KEAP1, NRF2, SLC7A11, and GPX4 in H1299 *KEAP1* KO cells were determined by western blotting. *KEAP1* KO cells were generated by *KEAP1* single guide RNA (sgRNA) infection. Vinculin was used as a loading control. sg sgRNA, C control. **c–f** Cell death upon erastin (**c**), RSL3 (**d**), and FIN56 (**e**) treatment in H1299 *KEAP1* KO cells was analyzed by PI staining and cell viability for FIN56 was estimated by CCK8 (**f**). Ctrl control. **g, h** Lipid peroxidation levels were determined for H1299 cells treated with RSL3 (**g**) and FIN56 (**h**). **i**, Protein levels of KEAP1, NRF2, SLC7A11 in H1299 *SLC7A11* KO, *KEAP1* KO, and *SLC7A11-KEAP1* DKO cells. **j, k** Cell death was quantified by PI staining for H1299 *SLC7A11* KO (*SLC* sg), *KEAP1* KO, and *SLC7A11-KEAP1* DKO (*KEAP1* sg + *SLC* sg) cells by RSL3 (**j**) and ML162 (**k**). **l, m** Lipid peroxidation levels were determined for H1299 *SLC7A11* KO, *KEAP1* KO, and *SLC7A11-KEAP1* DKO cells treated with RSL3 (**l**) and ML162 (**m**). **n** Protein levels of KEAP1, NRF2, GPX4 in H1299 *GPX4* KO, *KEAP1* KO, and *GPX4-KEAP1* DKO cells. **o–q** Cell death was quantified by PI staining upon ferrostatin-1 withdrawal for H1299 cells (**o**) and lipid peroxidation levels were determined (**p, q**). Data were presented as (if mentioned otherwise) mean ± SD; $n = 3$. *P* value was determined by two-way ANOVA; ns not significant. Source data are provided as a Source Data file.

Fig. 2e–h), but not by FIN56 (Supplementary Fig. 2i, j). Finally, analyses of the data from Cancer Therapeutics Response Portal[43] revealed that *KEAP1* mutant NSCLC cells exhibit more resistance to class 2 FINs (RSL3, ML162, and ML210), but not to FIN56, than do *KEAP1* WT NSCLC cells (Supplementary Fig. 2k–n). Together, our data reveal that *KEAP1* deficiency or mutation in lung cancer cells promotes ferroptosis resistance to class 2 FINs but does not affect ferroptosis sensitivity to the class 3 FIN.

**FSP1 underlies the differential effects of class 2 and 3 FINs in *KEAP1* deficient lung cancer cells.** Both class 2 and 3 FINs impinge on GPX4; their major difference lies in their differential effects on CoQ: class 3 but not class 2 FINs deplete CoQ, which can act as a radical-trapping antioxidant with anti-ferroptosis activities upon its conversion to the reduced form $CoQH_2$[37]. On the basis of this, our aforementioned data suggest a hypothesis that *KEAP1* deficiency likely upregulates another anti-ferroptosis pathway that operates in parallel to GPX4 and in a $CoQH_2$-dependent manner (therefore, when CoQ is depleted by FIN56, the ferroptosis-suppressing effect afforded by the upregulation of this pathway in *KEAP1* KO cells is abolished, explaining why *KEAP1* deficiency does not affect ferroptosis sensitivity to FIN56). To test this hypothesis, we sought to identify a downstream effector of KEAP1 involved in the CoQ pathway in lung cancer. To this end, we analyzed the Cancer Genome Atlas (TCGA) LUAD dataset[31] and identified a list of genes that are significantly upregulated in *KEAP1* mutant LUADs compared with *KEAP1* WT ones. We then performed a Gene Ontology analysis on the overexpressed genes to identify enriched pathways, which revealed that 12 of the overexpressed genes are involved in the CoQ metabolic process, with *FSP1* being the most significantly upregulated one in *KEAP1* mutant LUADs (Fig. 2a, Supplementary Fig. 3a, and Supplementary Table 1).

FSP1 was recently identified as a potent ferroptosis suppressor that operates independently of GPX4 by reducing CoQ to $CoQH_2$ on the plasma membrane[9,10]. We, therefore, hypothesized that *KEAP1* deficiency might enhance RSL3 resistance by promoting FSP1 expression or activity. In support of this, we found that *KEAP1* deletion resulted in increased expression of FSP1 (but not DHODH, another enzyme involved in the CoQ pathway to regulate ferroptosis) in both H1299 and H23 cells (Fig. 2b and Supplementary Fig. 3b). Restoration of patient-derived KEAP1 G333C mutant in *KEAP1* KO H1299 cells failed to exert any rescuing effect on FSP1 levels, or erastin- or RSL3-induced ferroptosis (Supplementary Fig. 3c, d). Conversely, overexpression of KEAP1 in H460 cells decreased FSP1 expression (Supplementary Fig. 3e, f). Further analyses revealed that *FSP1*

expression is significantly upregulated in *KEAP1* mutant lung cancers (including both LUADs and LUSCs) compared with *KEAP1* WT cancers (Supplementary Fig. 3g, h). Although *FSP1* was initially proposed as a p53 transcriptional target[44], subsequent studies showed that p53 activation does not regulate FSP1 expression[10]. We found that *FSP1* expression did not correlate with *p53* status in LUADs (Supplementary Fig. 3i). *FSP1* expression correlated with *STK11/LKB1* status (Supplementary Fig. 3j), likely because *KEAP1* and *STK11* are frequently co-mutated in LUADs[45–47]. Analyses of Cancer Cell Line Encyclopedia data also revealed that *KEAP1* mutant NSCLC cells generally exhibit higher expression of *FSP1* than do *KEAP1* WT ones (Supplementary Fig. 3k). A recent study showed that FSP1 inhibitor (iFSP1) significantly sensitized some lung cancer cells, but not others, to RSL3[10]. Correlation analyses of RSL3 EC50 change by iFSP1 with *KEAP1* status in this panel of lung cancer cell lines revealed that iFSP1 exhibited more sensitizing effects to RSL3 in *KEAP1* WT lung cancer cells than in *KEAP1* mutant ones (Supplementary Fig. 3l), which correlates with higher expression levels of *FSP1* in *KEAP1* mutant lung cancer cells (Supplementary Fig. 3k).

Extending from these correlative analyses, we further showed that reducing *FSP1* expression in *KEAP1* KO H1299 cells (to the level similar to that in control cells) re-sensitized *KEAP1* deficient cells to RSL3 or ML162 (Fig. 2c–e). As expected, overexpression of FSP1 drove ferroptosis resistance (Supplementary Fig. 3m–o), whereas its deficiency promoted ferroptosis sensitivity in H1299 cells (Supplementary Fig. 3p–r). In both H1299 and H23 cells, *KEAP1* deficiency-induced resistance to RSL3 or ML162 was largely abolished under iFSP1 treatment (Fig. 2f, g and Supplementary Fig. 3s–u).

FSP1 functions as an oxidoreductase to reduce CoQ to $CoQH_2$, thereby protecting cells against ferroptosis[9,10]. We found that *KEAP1* deficiency in H1299 cells decreased the $CoQ/CoQH_2$ ratio, whereas normalizing the FSP1 level in *KEAP1* KO cells back to that in control cells also restored the $CoQ/CoQH_2$ ratio (Fig. 2h). Importantly, *KEAP1* deficiency-induced ferroptosis resistance to RSL3 or ML162 in these lung cancer cells was largely abolished under CoQ synthesis blockade conditions (by 4-chlorobenzoic acid [4-CBA] treatment) (Fig. 2i–l and Supplementary Fig. 3v–x). Together, these results strongly suggest that *KEAP1* deficiency upregulates FSP1 levels and that the CoQ-FSP1 signaling axis plays an important role in mediating ferroptosis resistance to class 2 FINs in *KEAP1* deficient lung cancer cells.

**KEAP1 regulates FSP1 through NRF2-mediated transcription.** The aforementioned data prompted further mechanistic studies

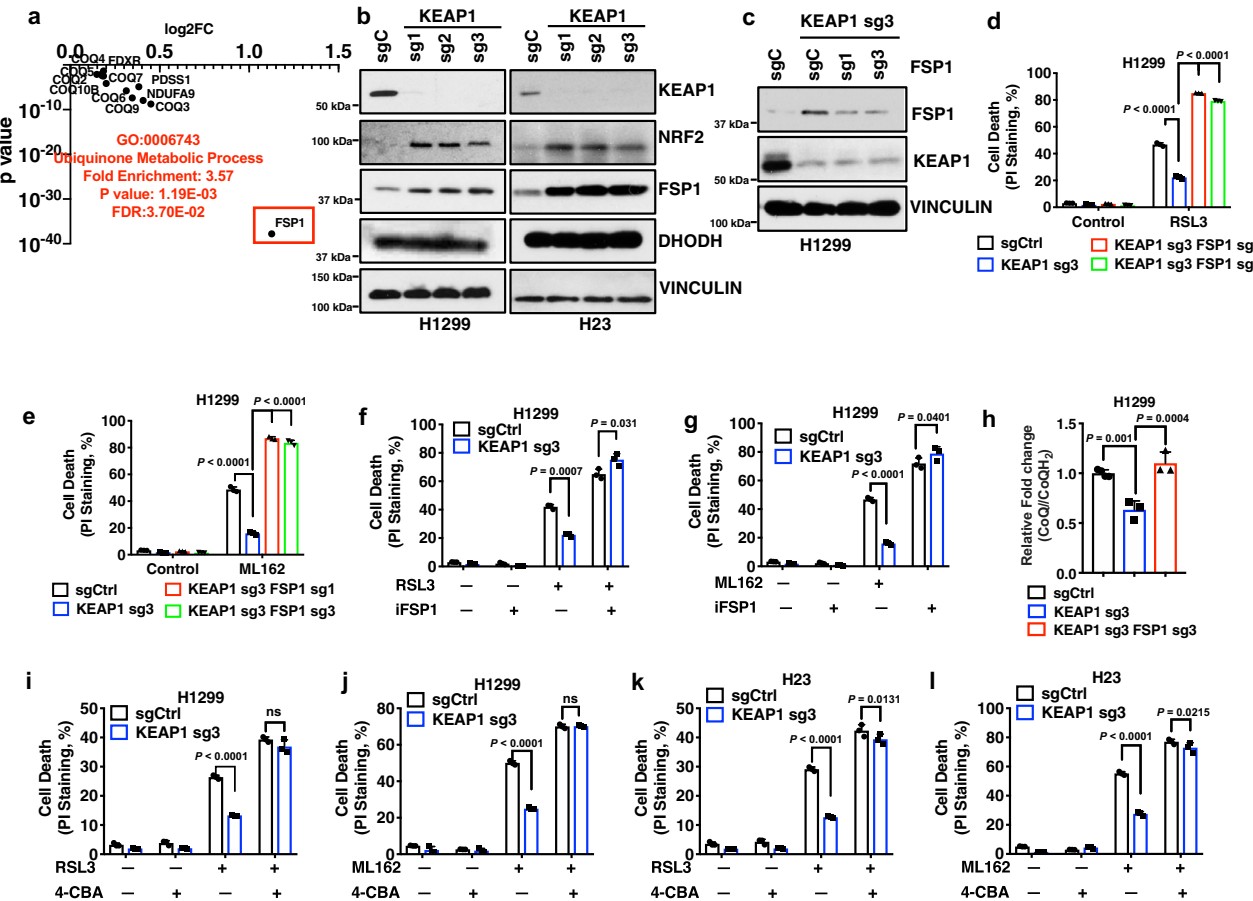

**Fig. 2 KEAP1 regulates ferroptosis sensitivity through FSP1. a** Gene ontology analysis revealed ubiquinol synthesis pathway enrichment in *KEAP1* mutant LUAD tumors. **b** Western blot analysis of KEAP1, NRF2, DHODH, and FSP1 protein levels in H1299 *KEAP1* KO and H23 *KEAP1* KO cells. **c** Western blot analysis of KEAP1, NRF2, and FSP1 protein levels in H1299 *KEAP1 FSP1* DKO cells. **d**, **e** Cell death analysis of H1299 *KEAP1 FSP1* DKO cells treated with RSL3 (**d**) and ML162 (**e**). **f**, **g** Cell death analysis of cotreatment of RSL3 (**f**) or ML162 (**g**) with iFSP1 in H1299 *KEAP1* KO cells. **h** Ubiquinone/ubiquinol (CoQ/CoQH₂) ratio in H1299 *KEAP1 FSP1* DKO cells. **i–l** Cell death analysis of H1299 *KEAP1* KO cells (**i**, **j**) and H23 *KEAP1* KO cells (**k**, **l**) cotreated with RSL3 + 4-CBA or ML162 + 4-CBA. Data were presented as (if mentioned otherwise) mean ± SD; *n* = 3. *P* value was determined by two-way ANOVA; ns not significant. Source data are provided as a Source Data file.

of how KEAP1 regulates FSP1 expression. Since *KEAP1* deficiency increased *FSP1* mRNA levels (Supplementary Fig. 3b) and considering that KEAP1 governs NRF2-mediated transcriptional programs[48], we examined whether *FSP1* is an NRF2 transcriptional target. Analyses of NRF2 chromatin immunoprecipitation coupled with high-throughput sequencing (ChIP-seq) datasets from GEO revealed a strong binding of NRF2 on the *FSP1/AIFM2* promoter in diverse cancer cell lines (Supplementary Fig. 4a, b; note that *FSP1* transcription is directed from the right to the left as indicated by the arrow). To further substantiate the link between FSP1 and NRF2 in cancers, we examined the correlations between *FSP1* expression levels and those of known NRF2 target genes in TCGA datasets. Unsupervised clustering analyses revealed positive correlations between the expression of *FSP1* and that of many NRF2 target genes in multiple cancers; notably, LUAD and LUSC exhibited the most striking correlations, wherein *FSP1* expression positively correlated with most of the NRF2 target genes (Supplementary Fig. 4c, d). These data, therefore, indicate a high relevance of FSP1 to NRF2 signaling in lung cancers.

NRF2 typically binds to antioxidant response elements (AREs) in gene promoter regions. Further analyses of the *FSP1* promoter identified two AREs located within 1 kb upstream of the transcription start site (TSS) of the *FSP1* gene

(Fig. 3a), which also overlapped with the NRF2 ChIP peak region (Supplementary Fig. 4b). ChIP analyses revealed strong NRF2 binding on both AREs in A549 cells (a *KEAP1* mutant lung cancer cell line) (Fig. 3b, c). Likewise, *KEAP1* deletion in H1299 cells significantly increased NRF2 binding on these AREs (Fig. 3d, e). Treatment with the NRF2 inducers tert-butylhydroquinone (TBHQ) increased NRF2 and FSP1 levels (Fig. 3f and Supplementary Fig. 4e, f), and correspondingly, significantly enhanced *FSP1* promoter-luciferase activities, which could be partially decreased by mutation of either ARE and almost completely abolished by mutation of both AREs (Fig. 3g, h). Likewise, *KEAP1* deletion increased *FSP1* promoter-luciferase activities and this increase was abolished by ARE mutations (Fig. 3i).

We further studied the relevance of NRF2 to KEAP1 regulation of FSP1 and ferroptosis. We found that *NRF2* deletion in *KEAP1* KO cells abolished *KEAP1* deficiency-induced *FSP1* promoter-luciferase activities and FSP1 expression (Fig. 3i, j and Supplementary Fig. 4g) and re-sensitized *KEAP1* KO cells to RSL3- or ML162-induced ferroptosis; importantly, FSP1 re-expression in *KEAP1/NRF2* double KO cells (to the level similar to that in *KEAP1* KO cells) restored ferroptosis resistance in these cells (Fig. 3k, l and Supplementary Fig. 4h). Together, these data suggest that *FSP1* is an NRF2 transcriptional target and that

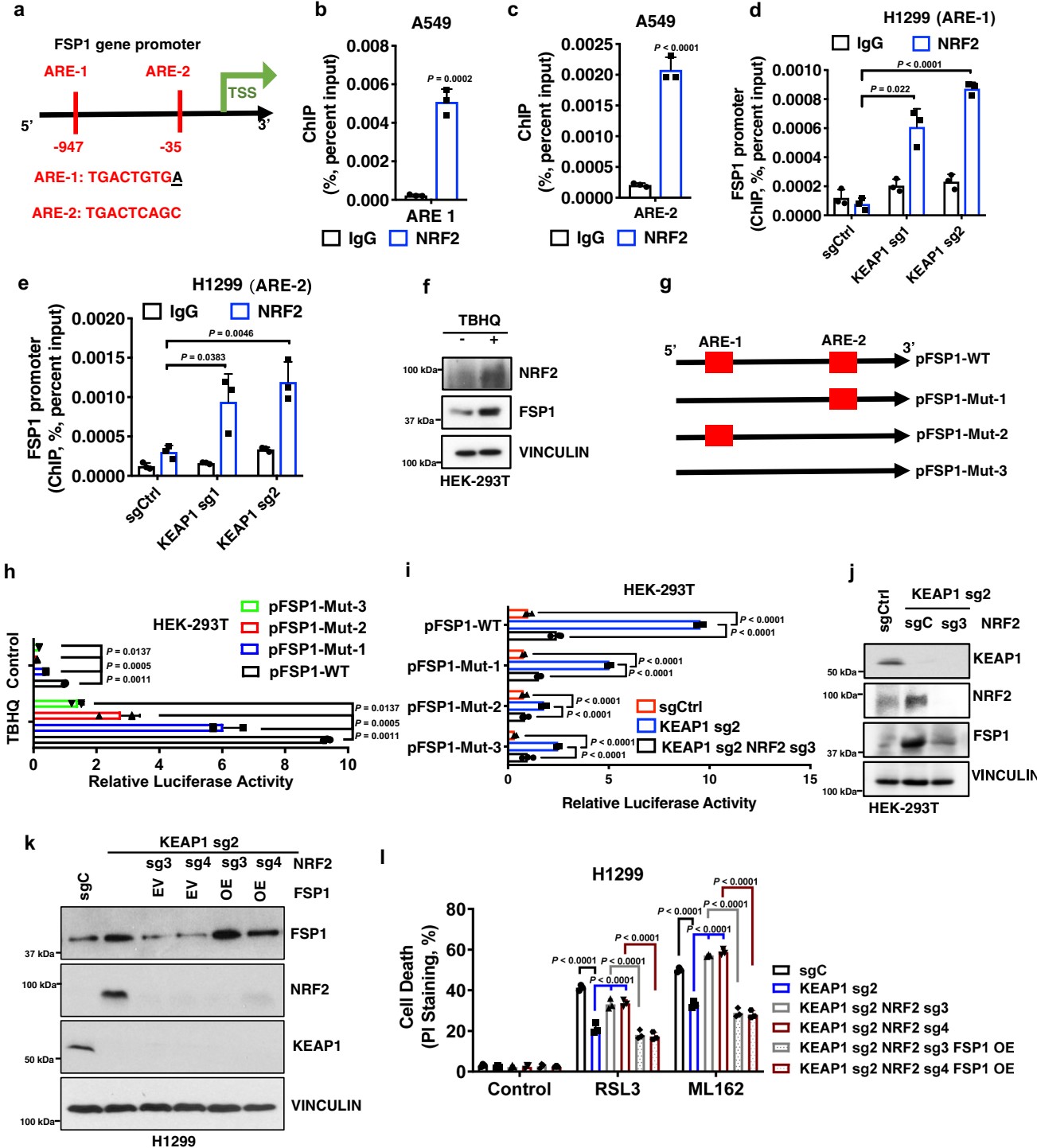

**Fig. 3 KEAP1 regulates FSP1 through NRF2-mediated transcription. a** Schematic showing ARE binding sites on the *FSP1* promoter region. **b**, **c** ChIP analyses of NRF2 binding on ARE1 (**b**) and ARE2 (**c**) in the *FSP1* promoter in A549 cells. **d**, **e** ChIP analyses of NRF2 binding on ARE1 (**d**) and ARE2 (**e**) in the *FSP1* promoter in H1299 *KEAP1* KO cells. **f** Protein levels of KEAP1, NRF2, FSP1 in HEK-293T cells treated with TBHQ. **g–j** *FSP1* promoter-luciferase activities in response to TBHQ treatment (**g**, **h**) or in *KEAP1 NRF2* DKO HEK-293T cells (**i**, **j**). **k** Protein levels of KEAP1, NRF2, and FSP1 in *KEAP1 NRF2* DKO H1299 cells overexpressing FSP1. EV empty vector, OE overexpression. **l** PI staining to measure cell death upon RSL3 or ML162 treatment in *KEAP1 NRF2* DKO H1299 cells overexpressing FSP1. Data were presented as (if mentioned otherwise) mean ± SD; n = 3. P value was determined by unpaired Student's t-test (3**b**–**e**) and two-way ANOVA (3**h**, **i**, **l**); ns not significant. Source data are provided as a Source Data file.

*KEAP1* deficiency in lung cancer cells leads to FSP1 upregulation through NRF2, resulting in ferroptosis resistance.

**FSP1 is critical for tumor growth in *KEAP1* deficient lung cancer.** Next, we studied the potential relevance of FSP1 to

human cancers through analysis of TCGA data. These analyses revealed that *FSP1* levels were upregulated in multiple cancers compared to their corresponding normal tissues, including LUAD, LUSC, kidney renal clear cell carcinoma (KIRC), kidney renal papillary cell carcinoma (KIRP), kidney chromophobe

(KICH), stomach adenocarcinoma (STAD), liver hepatocellular carcinoma (LIHC), and uterine corpus endometrial carcinoma (UCEC) (Fig. 4a). In addition, higher expression of *FSP1* correlated with shorter patient survival in LUAD, KIRC, and ovarian cancer (Fig. 4b).

We then investigated the role of FSP1 in *KEAP1* deficient lung tumor growth. While *KEAP1* and/or *FSP1* deletion did not obviously affect cell proliferation in H1299 cells (Supplementary Fig. 5a), loss of KEAP1 significantly accelerated H1299 xenograft tumor growth, which is consistent with previous reports[49,50]; importantly, *FSP1* deletion markedly suppressed *KEAP1* KO tumor growth (Fig. 4c, d). Staining of 4-hydroxy-2-noneal (4-HNE), a lipid peroxidation marker[51,52], in these tumor samples revealed that *KEAP1* deletion decreased 4-HNE staining, which was normalized by *FSP1* deletion in *KEAP1* deficient tumors (Fig. 4e, f). *KEAP1* deletion increased Ki67 staining whereas *FSP1* deletion did not affect Ki67 staining in *KEAP1* KO tumors (Supplementary Fig. 5b, c). Therefore, lipid peroxidation marker, but not cell proliferation marker, correlated with tumor growth in these tumors. Finally, we showed that FSP1 overexpression moderately promoted H1299 xenograft tumor growth and decreased 4-HNE staining in these tumors (Supplementary Fig. 5d–g). Taken together, our data show that, in the H1299 xenograft model, FSP1 is required (but probably not sufficient) for KEAP1 inactivation-induced tumor growth, and further suggest that FSP1 promotes *KEAP1* deficient lung tumor growth likely through suppressing lipid peroxidation and ferroptosis.

**Inhibiting FSP1 sensitizes *KEAP1* deficient lung cancer cells to radiation by inducing ferroptosis**. *KEAP1* mutant lung cancers are generally radioresistant[32,53]. We and others recently revealed an important role of ferroptosis in RT-induced cell death[21–24], prompting us to examine the potential role of FSP1 in mediating radioresistance in *KEAP1* deficient lung cancer cells. We confirmed that *KEAP1* deletion significantly promoted radioresistance in H1299 cells, and further showed that deleting *NRF2* re-sensitized *KEAP1* KO cells to RT, whereas re-expression of FSP1 in *KEAP1/NRF2* double KO cells restored radioresistance in these cells (Fig. 5a; Western blotting analyzing relevant protein expression in these cells is shown in Fig. 3k). *KEAP1* deficiency significantly attenuated RT-induced lipid peroxidation and ferroptosis marker gene *PTGS2* expression; this attenuation was abolished by *NRF2* deletion in *KEAP1* KO cells and then was restored by FSP1 re-expression in *KEAP1/NRF2* double KO cells (Fig. 5b, c). These data, therefore, established the NRF2-FSP1 axis as a key downstream effector to mediate radioresistance in *KEAP1* deficient lung cancer cells.

We further studied whether inactivating FSP1 sensitizes *KEAP1* deficient or mutant lung cancer cells to RT. We showed that genetically depleting *FSP1* sensitized *KEAP1* KO H1299 cells to RT (Fig. 5d), and restored RT-induced lipid peroxidation and *PTGS2* expression in *KEAP1* KO cells to a level similar to that in control cells (Fig. 5e, f). Likewise, iFSP1 treatment exerted a potent radiosensitizing effect (Fig. 5g) and promoted RT-induced lipid peroxidation in *KEAP1* deficient H1299 cells (Fig. 5h). Deleting *FSP1* also promoted radiosensitization and RT-induced lipid peroxidation in *KEAP1* mutant A549 cells (Fig. 5i–k). We obtained similar results by treating iFSP1 in several *KEAP1* mutant lung cancer cell lines (A549, H460, and H2126 cells) (Fig. 5l–o). Importantly, the radiosensitizing effect caused by *FSP1* deletion or iFSP1 treatment in *KEAP1* deficient H1299 cells could be largely abolished by the ferroptosis inhibitor ferrostatin-1 treatment (Supplementary Fig. 6). Collectively, these data suggest that FSP1 inactivation promotes radiosensitization in

*KEAP1* deficient or mutant lung cancer cells mainly through inducing ferroptosis.

**Inhibiting CoQ synthesis overcomes radioresistance in lung cancer cells or tumors with *KEAP1* deficiency or mutation**. Our aforementioned data suggest that FSP1 is an important therapeutic target and iFSP1 represents a promising radiosensitizer in treating *KEAP1* mutant cancer. However, iFSP1 cannot be used for in vivo treatment[10]. Since FSP1 operates in the same pathway with CoQ to suppress ferroptosis[9,10], we further tested whether inhibiting CoQ biosynthesis has any effect on radiosensitivity in lung cancer cells with KEAP1 inactivation. We showed that 4-CBA treatment reversed the radioresistance caused by *KEAP1* deletion (Fig. 6a), and restored RT-induced lipid peroxidation in *KEAP1* KO H1299 cells (Fig. 6b). 4-CBA treatment exerted similar radiosensitizing effects in *KEAP1* mutant A549 and H460 cells (Fig. 6c, d). These results were further corroborated by the genetic deletion of *COQ2* (which encodes a key enzyme involved in CoQ biosynthesis) in A549 and H460 cells (Fig. 6e–g). Importantly, 4-CBA treatment did not exert a radiosensitizing effect or promote RT-induced lipid peroxidation in human bronchial epithelial cells (HBECs) (Fig. 6h, i); therefore, 4-CBA appears to have a more potent ferroptosis-inducing or radiosensitizing effects in *KEAP1* inactivated lung cancer cells than in normal lung epithelial cells, suggesting a therapeutic window for the combination therapy with 4-CBA and RT.

We then investigated the therapeutic potential of combining 4-CBA treatment with RT for treating *KEAP1* mutant tumor growth. In the A549 xenograft tumor model, we showed that 4-CBA treatment moderately inhibited tumor growth, and combining 4-CBA with RT markedly suppressed tumor growth (Fig. 6j, k), which correlated with increased 4-HNE staining (but not with phospho-H2AX or cleaved caspase-3 staining) in these tumor samples (Fig. 6l, m and Supplementary Fig. 7a–d). We made similar observations in lung cancer patient-derived xenografts (PDXs) with *KEAP1* mutation (TC494) (Fig. 6n–q and Supplementary Fig. 7e–h). It should be noted that RT and 4-CBA treatment alone or in combination did not affect animal weight (Supplementary Fig. 7i, j), suggesting that these treatments were well-tolerated in animals. Collectively, our data suggest using of 4-CBA to overcome radioresistance in KEAP1 inactivated lung cancers.

## Discussion
The SLC7A11-GSH-GPX4 signaling axis constitutes the major surveillance system to defend against ferroptosis in cancer cells. Notably, multiple components in this signaling axis are well-established NRF2 transcriptional targets, including SLC7A11, catalytic and regulatory subunits of glutamate-cysteine ligases (GCLC and GCLM), glutathione synthetase (GS), and glutathione reductase (GR)[29,38]. In this study, we reveal that FSP1, another key arm in ferroptosis surveillance, is also an NRF2 transcriptional target, and its expression is governed by the KEAP1-NRF2 pathway; consequently, FSP1 expression is upregulated in *KEAP1* mutant lung cancers (Fig. 7a, b). Together, our study proposes that KEAP1-NRF2 controls both the SLC7A11-GSH-GPX4 and CoQ-FSP1 arms for ferroptosis defense; consequently, if one arm is disabled, *KEAP1* deficient cancer cells can still utilize the other arm to defend against ferroptosis. This model is supported by our observations (i) that *KEAP1* deficiency promotes ferroptosis resistance to class 1 or 2 FINs that inhibit SLC7A11 or GPX4, but does not affect ferroptosis induced by class 3 FIN (FIN56, which depletes GPX4 and CoQ and thereby inactivates both arms), and (ii) that GPX4 is essential in *KEAP1* WT H1299 cells, but not in *KEAP1* KO counterparts (likely because the increased levels of

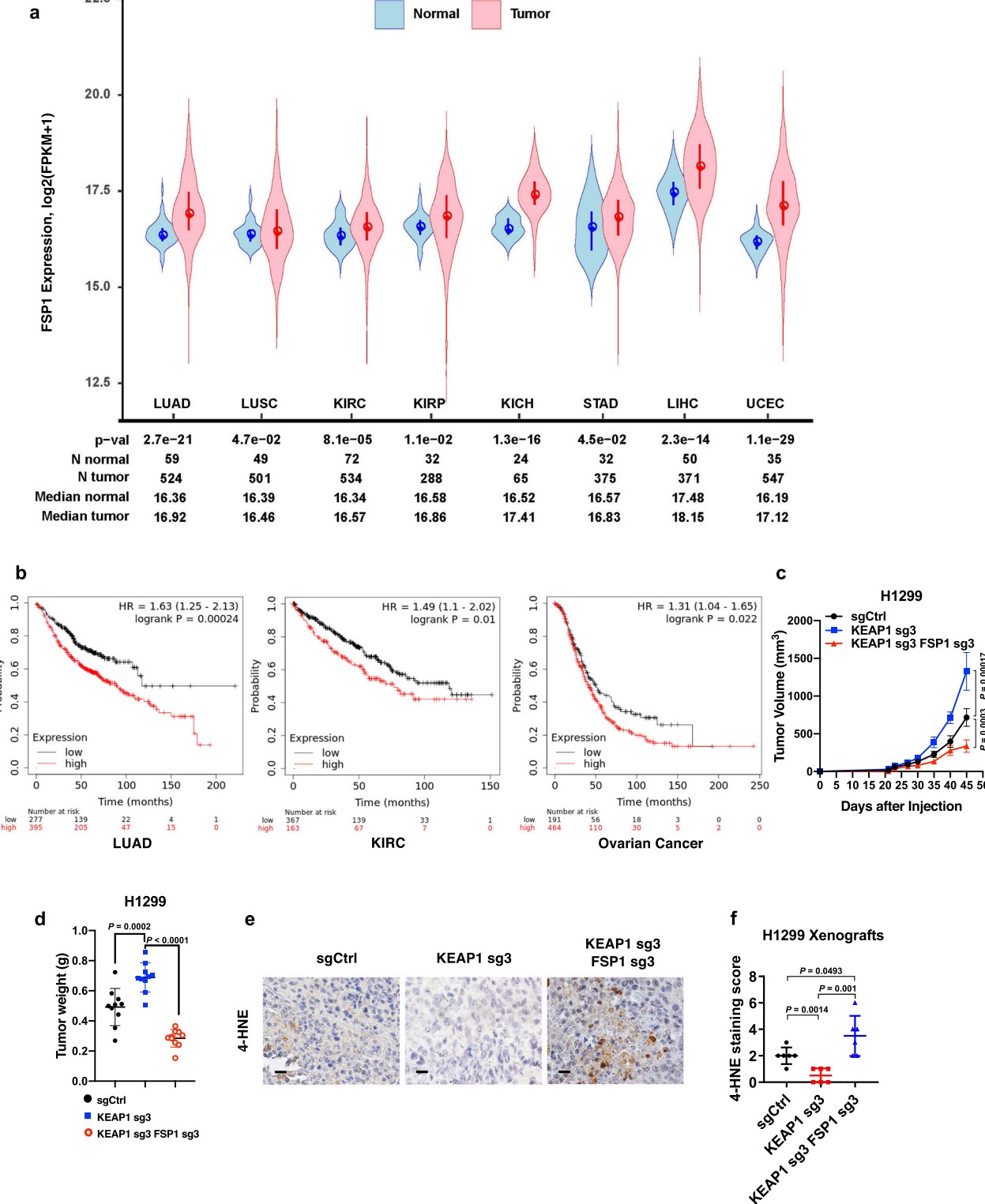

**Fig. 4 FSP1 promotes tumorigenesis in *KEAP1* deficient lung cancer. a** *FSP1* expression in tumor samples of lung adenocarcinoma (LUAD), lung squamous cell carcinoma (LUSC), kidney renal clear cell carcinoma (KIRC), kidney chromophobe (KICH), liver hepatocellular carcinoma (LIHC), uterine corpus endometrial carcinoma (UCEC), and stomach adenocarcinoma (STAD) vs corresponding normal tissues. **b** Survival analysis of various cancer types with high and low expression of *FSP1* in LUAD, KIRC, and ovarian cancer. **c, d** Measurement of tumor volumes (**c**) and endpoint tumor weights (**d**) of H1299 xenograft models with indicated genotypes. Error bars are means ± SD, *n* = 10 tumors. **e, f** Immunochemistry staining (**e**; scale bars, 20 μm) and scoring (**f**) of 4-HNE in H1299 xenograft tumors with indicated genotypes. Error bars are means ± SD, *n* = 6 randomly selected magnification fields. Data were presented as (if mentioned otherwise) mean ± SD; *n* = 3. For **d**, **f**, *P* value was determined by two-tailed unpaired Student's *t*-test; ns not significant. Source data are provided as a Source Data file.

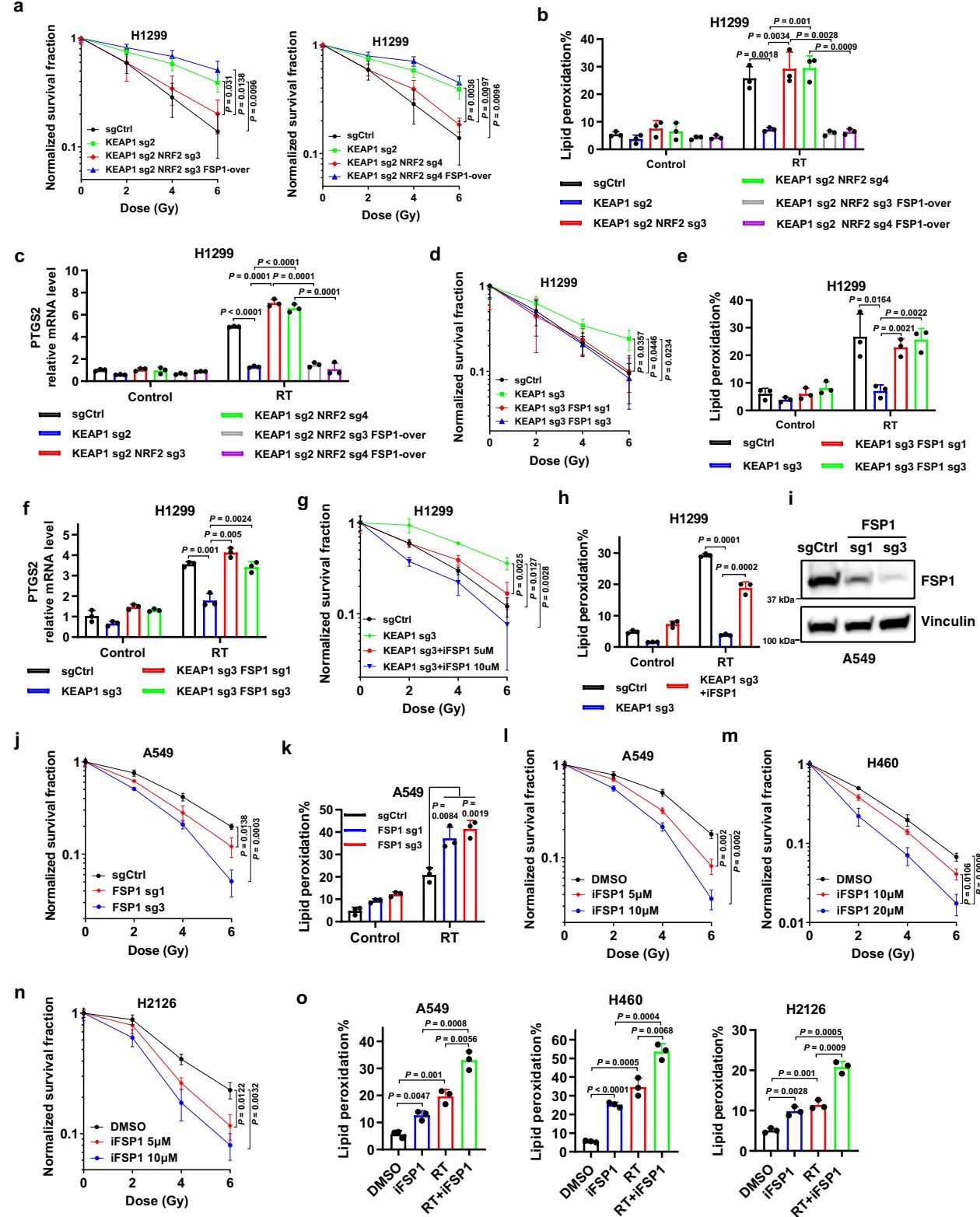

FSP1 can protect *KEAP1* KO cells against ferroptosis in the absence of GPX4). While other studies have also identified additional mechanisms mediating KEAP1-NRF2 function in ferroptosis governance in other cellular or cancer contexts[29,54–56], our study strongly suggests that, at least in lung cancer, FSP1 is a key effector to mediate ferroptosis resistance downstream of KEAP1-NRF2.

*KEAP1* deficiency increases GPX4 levels in H23 cells but decreased GPX4 expression in H1299 cells. Another recent study showed that *NRF2* knockdown increased GPX4 levels in some lung cancer cell lines[42]. The mechanisms underlying these discrepancies in different lung cancer cell lines remain unknown but could relate to other genetic factors that might have influenced NRF2 regulation of *GPX4* expression. Further study is required to

**Fig. 5 FSP1 inhibition sensitizes *KEAP1* deficient lung cancer cells to radiation by inducing ferroptosis. a** Clonogenic survival curves of the indicated H1299 cells exposed to X-ray irradiation at indicated doses. **b, c** Lipid peroxidation levels (**b**) and *PTGS2* mRNA levels (**c**) in the indicated H1299 cells at 24 h after 6 Gy X-ray irradiation. **d** Clonogenic survival curves of the indicated H1299 cells exposed to X-ray irradiation at indicated doses. **e, f** lipid peroxidation levels (**e**) and *PTGS2* mRNA levels (**f**) in the indicated H1299 cells at 24 h after 6 Gy X-ray irradiation. **g** Clonogenic survival curves of the indicated H1299 cells exposed to X-ray irradiation at indicated doses following pretreatment with the indicated doses of iFSP1 or DMSO for 24 h. **h** Lipid peroxidation levels in the indicated H1299 cells at 24 h after exposure to 6 Gy X-ray irradiation following pretreatment with iFSP1 or DMSO for 24 h. **i** Protein levels of FSP1 in A549 *FSP1* KO cells. **j** Clonogenic survival curves of the indicated A549 cells exposed to X-ray irradiation at indicated doses. **k** Lipid peroxidation levels in the indicated A549 cells at 24 h after exposure to 6 Gy X-ray irradiation. **l–n** Clonogenic survival curves of A549 (**l**), H460 (**m**), and H2126 (**n**) cells exposed to X-ray irradiation at indicated doses following pretreatment with the indicated doses of iFSP1 or DMSO for 24 h. **o** Lipid peroxidation levels in A549, H460, and H2126 cells at 24 h after exposure to 6 Gy X-ray irradiation following pretreatment with iFSP1 or DMSO for 24 h. Data were presented as (if mentioned otherwise) mean ± SD; $n = 3$. *P* value was determined by a two-tailed unpaired Student's *t*-test; ns not significant. Source data are provided as a Source Data file.

understand this context-dependent effect on GPX4 in lung cancer cells.

It should be noted that, while *Gpx4* is an essential gene in mouse[57], *Fsp1* (which was named *Amid* in this earlier study) is not required for mouse development[58]. From a therapeutic perspective, FSP1 appears to be a better therapeutic target than GPX4 in cancer treatment, as inhibiting FSP1 presumably would cause less toxicities in normal tissues than would inhibiting GPX4. In this study, we showed that FSP1 is overexpressed in *KEAP1* mutant lung cancers and that FSP1 is required for xenograft tumor growth in *KEAP1* deficient lung cancers, further nominating FSP1 as a promising therapeutic target for treating *KEAP1* deficient lung cancers (Fig. 7c). Since iFSP1, the only currently available FSP1 inhibitor, cannot be used in animal treatment, we hope that our study can motivate the future development of potent FSP1 inhibitors suitable for in vivo treatment and further testing of such inhibitors as radiosensitizers in treating *KEAP1* mutant lung cancers. Our data also suggest that targeting CoQ synthesis (by 4-CBA treatment) might provide another strategy to overcome radioresistance in *KEAP1* mutant lung cancers (Fig. 7c). Considering that immunotherapy has also been shown to induce ferroptosis[59], it will be interesting to further explore the combinations of 4-CBA and radiotherapy + immunotherapy in future preclinical investigations. Notably, 4-CBA treatment did not affect RT-induced lipid peroxidation and had no radiosensitizing effect in HBECs (Fig. 6h, i), suggesting that 4-CBA might selectively affect *KEAP1* mutant lung cancer cells while sparing normal lung epithelial cells. As far as we know, no drug targeting the CoQ-FSP1 pathway (including 4-CBA) has moved to clinical testing yet. We hope our current study can motivate the development of additional compounds that target this signaling axis for clinical application in the future.

FSP1 is an oxidoreductase that utilizes NADPH to reduce CoQ to CoQH$_2$[9,10]. Therefore, much like the GSH-GPX4 antioxidant system requiring NADPH to regenerate GSH (through GR), NADPH is also important for FSP1's function to regenerate CoQH$_2$ for ferroptosis defense. Notably, multiple NADPH-generating enzymes, including glucose-6-phosphate dehydrogenase (G6PD), 6-phosphogluconate dehydrogenase (PGD), isocitrate dehydrogenase 1 (IDH1), and malic enzyme 1 (ME1), are NRF2 targets[48]. This increased capacity to generate NADPH would be critical for buffering the increased NADPH consumption mediated by FSP1 to produce CoQH$_2$, and presumably also contribute to ferroptosis resistance in *KEAP1* deficient lung cancer cells, which provide additional insights on why these NADPH-generate enzymes are selected as NRF2 targets and are significantly upregulated in *KEAP1* mutant lung cancers[48]. In support of this hypothesis, we found that *FSP1* deletion in A549 cells decreased the NADP$^+$/NADPH ratio (Supplementary Fig. 8a), whereas FSP1 overexpression in H1299 cells had the opposite effect (Supplementary Fig. 8b).

In summary, through understanding the mechanistic underpinnings of ferroptosis phenotypes induced by different classes of FINs in *KEAP1* deficient lung cancer cells, in this study we identified the CoQ-FSP1 axis as a key downstream effector of the KEAP1-NRF2 pathway to mediate ferroptosis- and radiation-resistance in *KEAP1* deficient lung cancers. We further propose that pharmacological targeting of CoQ-FSP1 signaling can be exploited to overcome *KEAP1* deficiency-induced radioresistance and to treat *KEAP1* mutant lung cancers. Our study, therefore, identifies a potentially effective therapeutic strategy for treating this deadly disease.

## Methods

**Cell culture studies.** H1299, H23, H460, H2126, and A549 were cultured in a 37 °C incubator in a 5% CO$_2$ atmosphere. A549 and H2126 cells were cultured in Dulbecco's modified Eagle medium supplemented with 10% fetal bovine serum and 10,000 U/mL of penicillin-streptomycin. H460, H1299, and H23 cells were cultured in RPMI-1640 medium supplemented with 10% fetal bovine serum and 10,000 U/mL of penicillin-streptomycin. See Table 1 for detailed information on cell lines (as well as other reagents) used in this study.

**Constructs and generation of overexpression, knockdown, or knockout cell lines.** CRISPR-mediated knockout plasmids containing guide RNAs targeting *KEAP1*, *NRF2*, *FSP1*, *GPX4*, *COQ2*, and *SLC7A11* were generated in LentiCRISPR-V2 (Addgene, #52961). The expression vectors containing *FSP1* cDNA was used to generate stable cell lines as previously reported[11,60]. The *KEAP1* cDNA was described in our recent publication[50]. The *KEAP1* cDNA was subsequently cloned into the lentivirus expression vector pLVX-Puromycin and transfection was performed to generate stable cell lines as described previously[61]. The mutant *KEAP1* cDNAs were generated by the PCR mutagenesis using a Q5® Site-Directed Mutagenesis Kit (NEB) for DNA base substitutions according to the manufacturer's instructions. The mutant *KEAP1* cDNAs were cloned into the lentivirus vector pLX304-Blasticidin with an N-terminal V5 tag and transfection was performed to generate stable cell lines. The sequences of all the used primers are listed in Supplementary Table 2.

**Real-time PCR.** Real-time PCR was performed as described previously[62,63]. Briefly, an RNeasy kit (Qiagen) was used to extract total RNA from cells, and a high capacity cDNA reverse transcription kit (Applied Biosystems, ABI) was used to prepare first-strand cDNA. Subsequently, QuantiTect SYBR Green PCR kit (Qiagen) or TaqMan Universal PCR Master Mix (ABI) was used to perform real-time PCR on Stratagene MX3000P. For quantification of gene expression, the $2^{-\Delta\Delta Ct}$ method was used with expression normalized to β-Actin. The sequences of all the primers are listed in Supplementary Table 2.

**Cell death and viability assays.** Cell death or cell viability was measured as previously described[64,65]. Briefly, cells were seeded into 12-well plates or 96-well plates 24 h before treatment. Cell death after relevant treatments was measured by PI staining followed by flow cytometry analysis. Trypsinized cells were incubated in 100 µl of PBS containing 2 µg/ml of PI for 30 min at room temperature and analyzed with an Accuri 6 cytometer (BD Bioscience). Cell viability was measured using the CCK8 kit (Doijindo, Japan). Cells were seeded into 96-well plates. Following appropriate treatment, cell culture media was replaced with media containing 10% CCK8 reagent. The plate was incubated for 1 h at 37 °C in the CO$_2$ incubator and the OD value at 450 nm was measured by a microplate reader (FLUOstar Omega, BMG Labtech) to quantify cell viability.

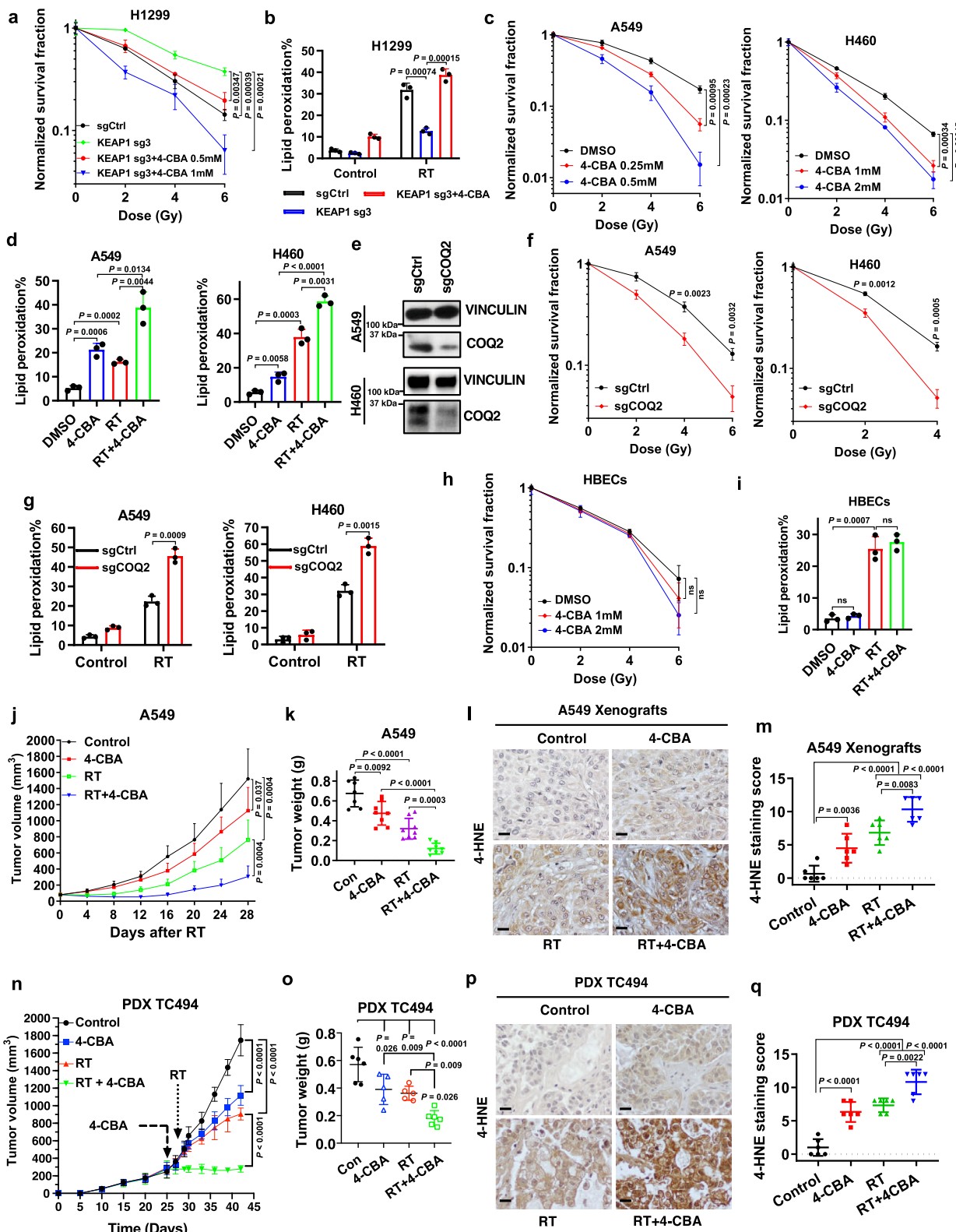

## Western blotting

Western blotting was conducted as previously described[66,67]. Cell lysates were prepared in NP40 buffer as previously described[68]. The primary antibodies and concentrations used for Western blotting were following: KEAP1 (Santa Cruz, sc-365626, 1:1000 dilution), FSP1 (1:1,000, Proteintech, no. 20886-1-AP), vinculin (1:5,000, Sigma, no. V4505), SLC7A11 (1:2,000, Cell Signaling Technology, no. 12691 S), GPX4 (1:1,000, R&D Systems, no. MAB5457), COQ2 (1:1,000, Santa Cruz, no. sc-51707), NRF2 (1:1,000, Cell Signaling Technology, no. 12721 S), DHODH (1:1,000, Proteintech, no. 14877-1-AP).

## Lipid peroxidation assay

Lipid peroxidation levels were measured as previously described[69]. Briefly, cells were seeded in triplicate in 12-well plates 1 day before treatment, pretreated with or without drugs for 24 h, and/or then irradiated. After the cells were incubated for 24 or 48 h, the cell culture medium of each well was replaced with a fresh medium containing 5 μM BODIPY 581/591 C11 dye (Invitrogen, D3861) for lipid peroxidation measurements and incubated for 30 min in a humidified incubator (at 37 °C, 5% CO$_2$). Subsequently, cells were washed with PBS and trypsinized to obtain a cell suspension. Lipid peroxidation levels were

 11

**Fig. 6 Inhibiting CoQ synthesis reverses radioresistance in *KEAP1* deficient or mutant lung cancer cells or tumors. a** Clonogenic survival curves of the indicated H1299 cells exposed to X-ray irradiation at indicated doses following pretreatment with the indicated doses of 4-CBA or DMSO for 24 h. **b** Lipid peroxidation levels in the indicated H1299 cells at 24 h after exposure to 6 Gy X-ray irradiation following pretreatment with 4-CBA or DMSO for 24 h. **c** Clonogenic survival curves of A549 and H460 cells exposed to X-ray irradiation at indicated doses following pretreatment with the indicated doses of 4-CBA or DMSO for 24 h. **d** Lipid peroxidation levels in A549 and H460 cells at 24 h after exposure to 6 Gy X-ray irradiation following pretreatment with 4-CBA or DMSO for 24 h. **e** Protein levels of COQ$_2$ in A549 and H460 *COQ$_2$* KO cells. **f** Clonogenic survival curve of the indicated A549 and H460 cells exposed to X-ray irradiation at indicated doses. **g** Lipid peroxidation levels in the indicated A549 or H460 cells at 24 h after exposure to 6 Gy X-ray irradiation. **h** Clonogenic survival curves of HBECs cells exposed to X-ray irradiation at indicated doses following pretreatment with the indicated doses of 4-CBA or DMSO for 24 h. **i** Lipid peroxidation levels in HBECs cells at 24 h after exposure to 6 Gy X-ray irradiation following pretreatment with 4-CBA or DMSO for 24 h. **j** Tumor volumes of A549 xenografts with indicated treatments at different time points (days) following exposure to 10 Gy of X-ray irradiation. Error bars are means ± SD, $n = 7$ or 8 tumors. **k** Tumor weights of A549 xenografts in the indicated treatment groups. Error bars are means ± SD, $n = 7$ or 8 tumors. **l, m** Representative images (**l**; scale bars, 20 μm) and scores (**m**) of IHC staining for 4-HNE in A549 xenograft tumors with indicated treatments. Error bars are means ± SD, $n = 6$ randomly selected magnification fields. **n** Tumor volumes of PDX TC494 in the indicated treatment groups at different time points (days) following exposure to 10 Gy of X-ray irradiation. Error bars are means ± SD, $n = 5$ or 6 tumors. **o** Tumor weights of PDX TC494 in the indicated treatment groups. Error bars are means ± SD, $n = 5$ or 6 tumors. **p, q** Representative images (**p**; scale bars, 20 μm) and scores (**q**) of IHC staining for 4-HNE in PDX TC494 tumors with indicated treatments. Error bars are means ± SD, $n = 6$ randomly selected magnification fields. Data were presented as (if mentioned otherwise) mean ± SD; $n = 3$. *P* value was determined by a two-tailed unpaired Student's *t*-test; ns not significant. Source data are provided as a Source Data file.

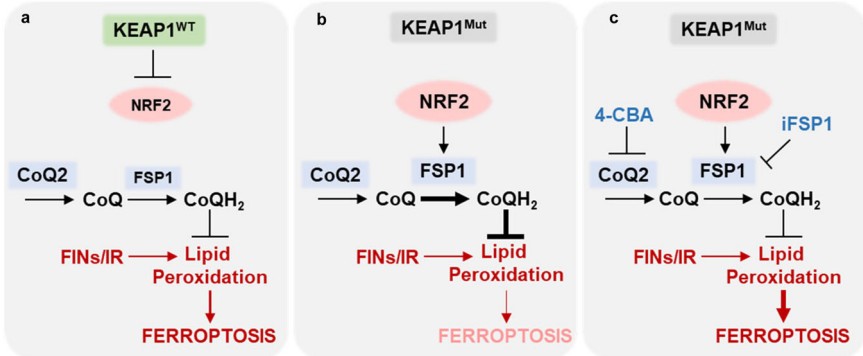

**Fig. 7 The working model depicting that FSP1 as an NRF2 transcriptional target is governed by the KEAP1-NRF2 pathway (a, b) and that a targetable CoQ-FSP1 axis drives ferroptosis- and radiation-resistance in KEAP1 inactive lung cancers (b, c).** See Discussion for a detailed description.

analyzed by flow cytometry using an Accuri 6 cytometer (BD Bioscience). The gating strategy used for the assay was shown in Supplementary Fig. 9.

**CoQ and CoQH$_2$ analysis.** Ubiquinone (CoQ) and ubiquinol (CoQH$_2$) were measured as described in our previous publication[11]. Specifically, cells were grown to 70% confluency in 35 mm. On the day of the extraction, the cells were quickly washed once in 1 ml of room temperature, serum-free culture medium to remove serum-derived CoQ and CoQH$_2$ with minimal metabolic perturbation to the cells. Cell extracts were prepared in 600 μl freshly prepared, ice-cold isopropanol containing 100 μM *tert*-butyl-hydroquinone to the plates under ice-cold conditions. The cells were scraped and the cell suspension was transferred to Eppendorf tubes on ice. After brief vortexing, the tubes were centrifuged at 13,000 RCF at 4 °C for 5 min and the supernatants were transferred to fresh tubes. These extracts were stored on dry ice (up to 48 h) in the dark until analysis by liquid chromatography with mass spectrometry (LC–MS).

CoQ and CoQH$_2$ levels were determined by LC–MS using an analytical method based on a lipidomics method previously described[70]. The extracts were transferred to amber glass sample vials in an autosampler maintained at 4 °C. Separation was performed on a Thermo Scientific UltiMate 3000 HPLC system using an Acquity UPLC HSS T3 column (2.1 × 100 mm, 1.8 μm particle size). The HPLC parameters are as follows: Mobile phase A (acetonitrile:water (60:40, v/v) with 10 mM ammonium acetate and 0.1% acetic acid), Mobile phase B (isopropanol:acetonitrile:water (85:10:5, v/v/v) with 10 mM ammonium acetate and 0.1% acetic acid), Gradient (0 min, 40% B; 1.5 min, 40% B; 12 min, 100% B; 15 min, 100% B; 16 min, 40% B; 17 min, 40% B), Injection volume (10 μl), Column temperature (55 °C), and Flow rate (400 μl/min).

Samples were analysed on an Exactive orbitrap mass spectrometer in positive ionization mode with a heated electrospray ion source. The instrument parameters are as follows: sheath gas flow rate 30 (arbitrary units), aux gas flow rate 10 (arbitrary units), sweep gas flow rate 3 (arbitrary units), spray voltage 4 kV, capillary temperature 120 °C, heater temperature 500 °C, capillary voltage 65 V, tube lens voltage 100 V. The scan range was set to 200–1000 *m/z*, with a maximum inject time of 100 ms, resolution of 100,000 at 1 Hz, and AGC (automatic gain control) target $1 \times 10^6$. The data were analysed using the MAVEN software suite 57

with signal intensity determined as the Peak Area (Top). Both CoQ and CoQH$_2$ were detected as their ammonium adducts ([M + NH4]$^+$).

**NADP$^+$/NADPH measurement.** NADP$^+$ and NADPH were measured as described in our previous publication[71]. Briefly, cells were grown to appropriate confluency in six-well plates. On the day of the extraction, the cells were lysed in a 300 μl extraction buffer (20 mM nicotinamide, 20 mM NaHCO$_3$, and 100 mM Na$_2$CO$_3$), followed by centrifugation and separation of the supernatant into two 150 μl aliquots. For total NADP measurements, 20 μl of cell supernatant from a 150 μl aliquot was mixed with 80 μl of NADP-cycling buffer (100 mM Tris-HCl pH8.0, 0.5 mM thiazolyl blue, 2 mM phenazine ethosulfate, and 5 mM EDTA) containing 0.75 U of G6PD enzyme (Sigma, no. G4134) in a 96-well plate, followed by incubation for 1 min in the dark. Then 20 μl of 10 mM glucose-6-phosphate was mixed into the 96-well plate, followed by a measurement of the change in absorbance at 570 nm at 1-min intervals for 6 min using a microplate reader. For NADPH measurement, the other 150 μl supernatant was incubated at 60 °C for 30 min, followed by the same procedures in parallel as for total NADP measurement. Eventually, the concentration of NADP$^+$ was calculated by subtracting [NADPH] from [total NADP].

**Irradiation and clonogenic survival assay.** For all clonogenic survival assays, cells were irradiated with an X-RAD 320 cabinet irradiator (Precision X-Ray) at doses from 0 to 6 Gy and a dose rate of 250 MU/min. About 200–1000 cells/well were seeded in triplicates in six-well plates and allowed to grow from 24 h. To determine the synergistic effects of the drugs with IR, cells were pretreated with iFSP1 or 4-CBA for 24 h, then irradiated and cultured in a normal medium. Fresh medium with or without ferroptosis inhibitors was added to the plates every 48 h. After incubation for 1–2 weeks, cells were stained with 0.5% crystal violet (Sigma, #C0775) dissolved in 20% methanol. The colonies in each well were counted visually. The surviving fraction was calculated using GraphPad Prism 6 and normalized to that of unirradiated control cells. The surviving fraction was depicted on a logarithmic scale and plotted on the y-axis against doses on the x-axis.

**Table 1 Key resources table.**

| Reagent or resource | Source | Identifier |
|---|---|---|
| Antibodies | | |
| KEAP1 | Santa Cruz Biotechnology | Cat# sc-365626 |
| Vinculin | Sigma | Cat# V4505 |
| SLC7A11 | Cell Signaling Technology | Cat# 12691 S |
| GPX4 | R&D Systems | Cat# MAB5457 |
| FSP1 | Proteintech | Cat# 20886-1-AP |
| NRF2 | Cell Signaling Technology | Cat# 12721 S |
| COQ$_2$ | Santa Cruz Biotechnology | Cat# sc-517107 |
| DHODH | Proteintech | Cat# 14877-1-AP |
| anti-Ki-67 (D2H10) | Cell Signaling Technology | Cat# 9027 |
| anti-4-HNE | Abcam | Cat# ab46545 |
| anti-Phospho-Histone H2A.X (Ser139) | EMD Millipore | Cat# 05–636 |
| anti-Cleaved Caspase-3 | Cell Signaling Technology | Cat# 9661 |
| Biological Samples | | |
| PDX TC494 | University of Texas MD Anderson Cancer Center | N/A |
| Chemicals, Peptides, and Recombinant Proteins | | |
| RSL3 | Cayman Chemical | Cat# 19288 |
| ML162 | Cayman Chemical | Cat# 20455 |
| FIN56 | Selleck | Cat# S8254 |
| 4-CBA (4-chlorobenzoic acid) | Sigma | Cat# 135585 |
| iFSP1 | Cayman Chemical | Cat# 29483 |
| TBHQ | Selleck | Cat# S4990 |
| Apigenin | Cayman | Cat# 10010275 |
| SFN | Selleck | Cat# S5771 |
| Corn oil | Sigma | Cat# 8001-30-7 |
| Erastin | Selleck | Cat# S7247 |
| Ferrostatin | Sigma | Cat# SML0583 |
| BODIPY$^{TM}$ 581/591 C11 | ThermoFischer | Cat# D3861 |
| Propidium Iodide | Roche | Cat# 11348639001 |
| RPMI-1640 | Sigma | Cat# R8758 |
| Penicillin-Streptomycin | Life Technologies | Cat# 15140-122 |
| DMEM | Sigma | Cat# D6429 |
| DPBS | Sigma | Cat# D8537 |
| Critical Commercial Assays | | |
| cDNA Reverse Transcription Kit | Applied Biosystems | Cat# 43-688-14 |
| QuantiTect SYBR Green PCR Kit | Qiagen | Cat# 204143 |
| TaqMan™ Universal PCR Master Mix | Applied Biosystems | Cat# 4305719 |
| CCK8 | Dojindo | Cat# CK0413 |
| Experimental Models: Cell Lines | | |
| H1299 | Laboratory of Bingliang Fang | N/A |
| H23 | Laboratory of Bingliang Fang | N/A |
| H1703 | Laboratory of Bingliang Fang | N/A |
| H2126 | ATCC | Cat# CRL-5925 |
| H460 | Laboratory of Bingliang Fang | N/A |
| A549 | ATCC | Cat# CRL-7909 |
| HEK-293T | ATCC | CRL-11268 |
| Experimental Models: Organisms/Strains | | |
| Mouse: athymic nude mice (female, 4-6 weeks old) | Experimental Radiation Oncology Breeding Core Facility at MD Anderson Cancer Center | N/A |
| Mouse: NOD SCID Gamma (NSG) mice (female, 4-6 weeks old) | Experimental Radiation Oncology Breeding Core Facility at MD Anderson Cancer Center | N/A |
| Software and Algorithms | | |
| GraphPad | GraphPad | GraphPad |
| Cancer Dependency Map Portal | Cancer Dependency Map Portal | Cancer Dependency Map Portal |
| FlowJo | FlowJo | FlowJo |
| BD Accuri C6 Plus | BD Accuri C6 Plus | BD Accuri C6 Plus |
| UCSC Xena | UCSC Xena | UCSC Xena |

**Cell line xenograft experiments**. All the xenograft experiments were performed as previously described[72,73] and in accordance with a protocol approved by the Institutional Animal Care and Use Committee and Institutional Review Board at The University of Texas MD Anderson Cancer Center. The study is compliant with all relevant ethical regulations regarding animal research. Female 4- to 6- week-old athymic nude mice (Foxn1nu/Foxn1nu) were purchased from the Experimental Radiation Oncology Breeding Core Facility at MD Anderson Cancer Center and housed in the Animal Care Facility at the Department of Veterinary Medicine and Surgery at MD Anderson. Cancer cell lines were resuspended in FBS-free RPMI and the same number of cells were injected into mice subcutaneously. Tumor progression was monitored by bi-dimensional tumor measurements taken twice a week until the endpoint. The tumor volume was calculated according to the equation $volume = 0.5 \times length \times width^2$. When the tumors reached 50–100 mm$^3$ in volume, the mice were randomized into four groups and treated with PBS, 4-CBA, ionizing radiation, or 4-CBA plus ionizing radiation, respectively. Ionizing radiation (X-ray) was applied locally to the tumor in the flank of mice at 8 Gy.

4-CBA was dissolved in dimethyl sulfoxide (DMSO) and diluted in corn oil, then intraperitoneally injected into mice three times before irridation at a dose of 250 mg/kg followed by continued injection once every two days until the endpoint as indicated in the corresponding figures. The tumor volume was measured three times per week until the endpoint and calculated according to the equation volume $= 0.5 \times$ length $\times$ width$^2$. The maximal tumor (2 cm diameter) size/burden was not exceeded.

**Patient-derived xenograft (PDX) experiments**. PDXs were generated in accordance with protocols approved by the Institutional Review Board at The University of Texas MD Anderson Cancer Center. Informed consent was obtained from the patients and the study is compliant with all relevant ethical regulations regarding research involving human participants. Four- to six-week-old NOD SCID gamma (NSG) female mice were purchased from the Experimental Radiation Oncology Breeding Core Facility at MD Anderson Cancer Center and housed in the Animal Care Facility at the Department of Veterinary Medicine and Surgery at MD Anderson Cancer Center. The PDX model used in this study was originally obtained from the lung cancer PDX platform at MD Anderson Cancer Center. PDX experiments were performed as previously described[71]. Briefly, PDX tumors in cold DMEM media were minced into fragments 1–2 mm$^3$ in volume. Then each PDX tumor fragment was subcutaneously inoculated into the dorsal flank of NSG mice. When the tumors reached 50–100 mm$^3$ in volume, the mice were randomized into four groups and treated with PBS, 4-CBA, ionizing radiation, or 4-CBA plus ionizing radiation, respectively. Ionizing radiation (X-ray) was applied locally to the tumor in the flank of mice at 8 Gy. 4-CBA was dissolved in dimethyl sulfoxide (DMSO) and diluted in corn oil, then intraperitoneally injected into mice three times before irridation at a dose of 250 mg/kg followed by continued injection once every two days until the endpoint as indicated in the corresponding figures. The tumor volume was measured three times per week until the endpoint and calculated according to the equation volume $= 0.5 \times$ length $\times$ width$^2$. In our animal studies, the allowed maximal tumor size/burden (2 cm diameter) was not exceeded.

**Histology and immunohistochemistry**. Xenograft tumor and PDX tumor samples were collected and fixed in 10% neutral-buffered formalin (ThermoFisher Scientific) overnight. Tumors were washed with PBS and then transferred to 70% ethanol followed by embedding, sectioning, and hematoxylin and eosin staining. For immunohistochemical staining, tissue sections were processed as previously described[74,75]. The primary antibodies used for immunohistochemistry were anti-4-HNE (1:400, Abcam, ab46545), anti-phospho-histone H2A.X (1:500, EMD Millipore, 05–636), anti-Ki-67 (D2H10) (1:500, Cell Signaling Technology, 9027 S). Images were obtained at 400× magnification on an Olympus BX43 microscope.

**Gene expression correlation analysis in TCGA cancers and CCLE cell lines**. Data from the cancer dependency map[76] was downloaded from the Depmap data portal (https://depmap.org/portal/). Cell lines were selected based on their NSCLC classification. The cell lines were further classified based on their *KEAP1* mutation status. Data from the TCGA LUAD cohort were downloaded from the UCSC Xena browser and analyzed[77]. The expression data of *FSP1* (previously known as *AIFM2*) and NRF2 target genes, and patient survival data in 33 cancer types were obtained from TCGA. The differentially expressed genes in *KEAP1* mutant LUAD tumors were defined by a cutoff of FDR <0.05 with at least a 1.5-fold change. All these data were generated using the UCSC Xena Browser (http://xena.ucsc.edu/). Pearson's correlation (two-sided) analysis was used to determine the expression correlation between *FSP1* and NRF2 target genes. *P* values were calculated by correlation test and adjusted by Benjamini–Hochberg method.

**Survival analysis**. For the survival analysis with gene expression, survival was compared between two groups of patients separated by the expression levels of gene *FSP1*. The survival plots were generated from KM-plotter.

**Quantification, statistics, and reproducibility**. For knockout experiments, at least two independent sgRNAs were used to generate respective cell lines. In the case of overexpression experiments, lentivirus containing the gene of interest was prepared multiple times to verify overexpression efficiency in cells. Results of cell culture experiments were obtained from at least three independent repeats. Data were represented as means ± standard deviation (SD) calculated from $n = 3$. Gene set enrichment analysis illustrated in Supplementary Fig. 3a was performed with PANTHER enrichment analyses. For boxplots, the lower and upper edges of the box extend from the first and third quartiles (25th and 75th percentiles) of the data and the middle line represents the median. The whiskers extend from the minimum to the maximum. Statistical significance (*p* values) was calculated using two-sided unpaired Student's *t*-tests and two-way ANOVA analysis by GraphPad Prism 8.0 (GraphPad Software, Inc.). For all statistical analyses, the difference was considered significant with a *p* value <0.05. For immunoblots, the experiments have been repeated at least twice with similar results and representative data were shown.

**Reporting summary**. Further information on research design is available in the Nature Research Reporting Summary linked to this article.

## Data availability
The uncropped films for immunoblots used in this study are shown in Source Data files. All data were available within the paper and Source Data file. Source data are provided with this paper.

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

## Acknowledgements

This research was supported by the Institutional Research Fund and Bridge Fund from The University of Texas MD Anderson Cancer Center, Boot Walk Seed Award from Radiation Oncology Strategic Initiatives (ROSI) at The University of Texas MD Anderson Cancer Center, Cancer Prevention & Research Institute of Texas grant RP220258, and R01CA181196, R01CA244144, and R01CA247992 from the National Institutes of Health (to B.G.). B.G. was an Andrew Sabin Family Fellow. P.K. was supported by the CPRIT Research Training Grant (RP170067) and Dr. John J. Kopchick Research Award from The University of Texas MD Anderson Cancer Center UTHealth Graduate School of Biomedical Sciences. This research has also been supported by the National Institutes of Health Cancer Center Support Grant P30CA016672 (to The University of Texas MD Anderson Cancer Center).

## Author contributions

P.K. and G.L. contributed equally and performed most of the experiments with assistance from Y.Z., Y.Y., C.M., X.L., A.H., and M.D.; Y.Z. did ChIP analyses and established *FSP1* promoter-luciferase assay; K.O. and L.K. conducted CoQ and CoQH2 analyses; M.V.P. provided resources for CoQ and CoQH2 analyses. J.S. did some of the computational analyses under the guidance of W.L.; B.G., P.K., G.L., and Y.Z. designed the experiments; B.G. and P.K. wrote the manuscript; B.G. and G.L. did the revision. B.G. supervised the project, established the collaboration, and provided funding support for the project; all authors commented on the manuscript.

## Competing interests

K.O. is a full-time employee of the Barer Institute and a former full-time employee of Kadmon Corporation. L.K. and M.V.P. are full-time employees of Kadmon Corporation.
