## [Peer Review File · Nature Communications]

REVIEWER COMMENTS

Reviewer #1 (Remarks to the Author):

In the manuscript, Koppula and Lei et al. of the group of Boyi Gan present data on the CoQ-FSP1 axis that prevents ferroptosis and at the same time drives radiation resistance in lung cancer.

Ferroptosis is a recently discovered form of iron catalysed regulated necrosis (RN). In contrast to classical RN pathways such as necroptosis and pyroptosis, ferroptosis is not triggered by an active protein-mediated signalling but rather occurs if a failsafe, such as GPX4 or FSP1 cannot function properly. The two major anti-ferroptotic systems described so far include 1) the GPX4 system which requires glutathione (GSH) to prevent ferroptosis – it is inhibited by RSL3 and erastin. The second system, the FSP1 (AIFM2) oxidoreductase (as DHODH) reduce CoQ to CoQH2 which inhibits lipid peroxy radical formation as an RTA. Importantly, this system prevents ferroptosis in a GSH-independent manner.

The tumour suppressor KEAP1, through the transcription factor NRF2, regulates expression of hundreds of proteins, many of which are involved in redox biology control. It has been known for some time that KEAP1 is inactive in some cancers, including lung cancers.

Here, the authors established a link between ferroptosis and KEAP1-deficient lung cancer and suggest that pharmacological inhibition of the CoQ-FSP1 axis sensitizes these cancers to radiation. This also works if the cancer cells are injected into mice (PDX). The authors conclude that this targeting might represent a novel therapeutic strategy to target KEAP1 mutant lung cancer.

Clearly, this topic is of interest to the readership of Nature Communications, and the advance in knowledge provided by this work is valuable. That said, I cannot recommend publication in a journal such as Nature Communications in the present form without some major concerns listed below being adequately addressed. For further improvement of the manuscript, I listed some minor remarks.

Thank you for having me as a referee for Nature Communications.

Major concerns

- The standard deviations in all experiments that demonstrate PI staining are surprisingly low. Are these really independent repetitions or are the same results simply measured several times? Of course, they should be repetitions performed with independently seeded cells. As an example for one figure, this applies to Figures 1b, 1c, 1d, 1k, 1l, 1p, but needs to be justified for all figures. A very good example of this concern is Figure 3m. Maybe, to be more convincing in this respect, authors might chose to demonstrate original FACS plots for some individual experiments.
- 4-Carboxybenzaldehyd (4CBA) is a deriviate of benzaldehyd. Why would this by any means be specific for CoQ? Authos have to assess if 4CBA functions as a radical trapping agent which might explain the effects reported on the tumors in Fig. 6. Some cell lines do not express FSP1 and still do not die by ferroptosis spontaneously, but do so upon RSL3 or erastin treatment (Stockwell and Dixon data, Conrad data). Please also investigate the effects of 4CBA in settings that are independent of FSP1 to entirely rule out that 4CBA might inhibit GSH-dependent ferroptosis or ROS-induced cell death in general.
- "Lipid peroxidation" is an interpretation (e.g. Fig. 5e, 5h, 5o and other). The axis should be labelled with what was actually measured.
- It is currently under debate if ferroptosis is a pro- or an anti-inflammatory cell death (necroinflammatory potential of ferroptosis). Please discuss to which extent either hypothesis might influence the success of tumour rejection following radiation in your setting.

Minor remarks

- Why is the normalized survival fraction (e.g. Fig. 5a) demonstrated with a log axis?
- The list presented in Figure S1 is important and should and should not be hidden in the supplement of the manuscript.
- The sentence "Accumulating evidence indicates that ferroptosis is a critical tumor suppression 64 mechanism." requires direct citations.

Reviewer #2 (Remarks to the Author):

KEAP1 mutations are frequently observed in NSCLC and lead to resistant to most standard cancer therapy. In this manuscript, Gan group found that KEAP1 deletion significantly promotes ferroptosis resistance to RSL3 or ML162, which is independent of SLC7A11 or GPX4. The FSP1-CoQ10-NAD(P)H pathway exists as a stand-alone parallel system and cooperates with GPX4 and glutathione to suppress ferroptosis. Here, Gan group found that FSP1 is a NRF2 transcriptional target and that KEAP1 deficiency in lung cancer cells leads to FSP1 upregulation through NRF2, resulting in ferroptosis resistance. They further demonstrated that the NRF2-FSP1 axis acts as a key downstream effector to mediate radioresistance in KEAP1 deficient lung cancer cells. Finally, in addition to in vitro cell culture experiments, via in vivo xenograft and PDX mouse models, they found that blocking FSP1-CoQ10-NAD(P)H axis by targeting CoQ synthesis (by 4-CBA treatment) can sensitize KEAP1 mutant or deficient tumors to radiotherapy via inducing ferroptosis. This is well designed study with solid data to support the conclusion. Moreover, this study might provide another strategy to overcome radioresistance in KEAP1 mutant NSCLC, a deadly disease we are facing today. This manuscript is suitable for publication on Nature Communication.

Since FSP1-CoQ10 axis regulates NAD(P)H production, which plays an important role against lipid peroxidation. It would be great if that the alteration of NADPH level can be examined when targeting FSP1 or CoQ1 in in vitro cell culture experiments. In addition, it will interesting to know that whether 4-CBA or other compounds/chemicals that target FSP1-CoQ10-NAD(P)H axis to induce ferroptosis has been proposed or be in clinical trial for cancer treatment. This could be added in the discussion, which is essential to link current findings to translational cancer treatment.

Reviewer #3 (Remarks to the Author):

In this manuscript, the authors reported that FSP1 is an NRF2 target gene; and demonstrated that targeting FSP1 to include ferroptosis is a great strategy for treating KEAP1 mutant lung cancer. In particular, they found that inhibition of KEAP1 expression render lung cancer cells more resistance to class 2 FINs, but not to class 3 FIN, from which they identified two parallel pathways controlled by NRF2: (1) NRF2-SLC7A11-GPX4, and (2) NRF2-FSP-CoQ. Certainly this is a very significant and timely finding. Using H1299 xenograft model, the authors clearly demonstrated that knockout of FSP slowed down the growth of KEAP1-deficient tumors. Furthermore, using different lung cancer cell lines, the author demonstrated pharmacological and genetic deletion of FSP1 sensitizing cancer cells to radiation therapy (RT) in an NRF2-dependent manner. The translational value of this study was further demonstrated by the synergistic effects of using CBA (an inhibitor of COQ2 that can be used in vivo) and RT as a combination therapy. In addition, CBA radiosensitized lung cancer cells but not normal lung epithelial cells (HBECS). Finally a lung cancer PDX model with KEAP1 mutation was further used to demonstrate the effectiveness of RT and 4-CBA combination therapy. The quality of the data are high with appropriate statistics. The topic is well suited to broad readership of this journal.

The following minor points should be addressed.

“sensitivity analyses of KEAP1 deficient lung cancer cells to different ferroptosis inducers link KEAP1 function in ferroptosis regulation to ubiquinone (CoQ) pathways”

The authors used both “KEAP1 deficient lung cancer cells” or “KEAP1 inactivated lung cancer cells” throughout the entire manuscript. The H1299 and H23 cell lines used in this study have no mutation in KEAP1 or NRF2 based on this paper. However the authors deleted KEAP1 using sgRNA. So in that sense, “KEAP1 deficient” should be used. However, the in real world, we large see Keap1 mutated cancer cells, which would call for the use of “KEAP1 inactivated lung cancer cells”. So it is a tough call, how to label KEAP1-mutated cancer cells.

“the CoQ-FSP1 axis mediates ferroptosis- and radiation- resistance in KEAP1 deficient lung cancer cells”

The statement is not precise since the CoQ-FSP1 axis mediates ferroptosis- and radiation-resistance in any cells regardless the KEAP1 status. It is just the higher expression of FSP1 in cancer cells than normal cells make cancer cells more susceptible to FSP1 inhibition than normal cells.

“Oxidative stress or KEAP1 inactivation (by its loss-of-function mutation, deletion, or epigenetic silencing in cancers) displaces NRF2 from KEAP1 and stabilizes NRF2”

The two-side binding model of NRF2-KEAP1 does not support “displaces NRF2 from Keap1” model, since the strong binding site ETGE-Kelch is hard to be displaced.

“KEAP1 deletion in H1299 cells dramatically decreased GPX4 levels (Fig. 1A) in an NRF2-dependent manner, as deleting NRF2 in KEAP1 deficient H1299 cells restored GPX4 expression (Fig. S2B).”

It is interesting that NRF2 negatively regulating GPX4 levels in H1299, which is contradictory to the majority of papers published in the literature, reporting a positive correlation. This negative regulation of GPX4 by NRF2 should be discussed. Furthermore, it is intriguing to see that KEAP1 deletion increased GPX4 levels in H23 cells, whereas decreased GPX4 levels in H1299, however, their responses to all classes of FINs tested are very similar. Please explain?

“KEAP1 mutant H460 cells”

Please provide more information about the NRF2 and KEAP1 mutation status of H460.

“Together, these results strongly suggest that KEAP1 deficiency upregulates FSP1 levels and that the CoQ-FSP1 signaling axis plays an important role in mediating ferroptosis resistance to class I FINs in KEAP1 deficient lung cancer cells.”

I think “class II FINs” should be used here.

Although it would not affect the conclusions of this manuscript, but using apigenin as an NRF2 inducer is not a good idea since it affects so many pathways. Left panel of S Figure 3F should be either deleted or the experiment is repeated using sulforaphane as an NRF2 inducer.

“It should be noted that RT and 4-CBA treatment alone or in combination did not affect animal weight (Fig. S8E, F)”

I think the Y axis in S8F should be “Body weight (g)”.

Reviewer #4 (Remarks to the Author):

Radiotherapy (RT) is applied to treat more than half cancer patients would wide; whereas the exact mechanism underlying radiation mediated tumor cell death, although extensively studied, remains unelucidated. One of the questions is whether ferroptosis, a specific programmed cell death due to imbalanced lipid oxidation and digestion. Using KEAP1 inactivated lung cancer cells, this study by Koppula et al, revealed that KEAP1, a stress responsive protein, can regulate ferroptosis by activating CoQ pathway. This is evidenced by inhibition of ferroptosis by FSP1 indicating the CoQ-FSP1 axis. KEAP1 is shown to regulate ferroptosis suppressor protein 1 (FSP1) via NRF2 transcription suggests is found to be CoQ-FSP1 axis that plays a mediates ferroptosis- and radiation- resistance in KEAP1 deficient lung cancer cells. Importantly, inhibition of the CoQ-FSP1 axis demonstrates a potential therapeutic effect of radiosensitization of KEAP1 inactivated lung cancer cells and PDX which are all impressive. However, some gaps in the description of experimental design and data explanation weakened the conclusion. The precise KEAP1 mediated NRF1 regulation is to be clarified.

1. In Abstract, there are several gaps in describing the specific pathway. Why focusing on KEAP1 inactivated lung cancer is not justified. The ubiquinone (CoQ) is not logically linked to the study of NRF2. It is not clear how the CoQ-FSP1 axis is found. A cartoon illustrating the KEAP1 mediated ferroptosis may help to quickly see the innovative finding.

2. In Introduction, lung cancer patients with KEAP1 mutations have shorter overall survival with poor prognosis, and KEAP1 mutant lung tumors are refractory to most available standard-of-care therapies including RT, highlighting the pressing need to develop novel and effective combination therapies for this type of lung cancer. It is not clear what kinds of KEAP1 mutations are carried in these patients and whether the mutant forms are functional deficient or functional gained which is a fundamental issue for proposing the current study. Metadata analysis of lung cancer patient's outcome with different kind of KEAP1 mutants are to be illustrated.

3. Fig. 1 is to reveal KEAP1 specifically involved in ferroptosis, thus other kinds of cell death should be considered as controls. In Fig1 legend, sg1 sg2 etc shown in the panel are not explained. In addition, in this figure, KEAP1 is studied in only one H1299 KEAP1 KO cell. At least another lung cancer cell line as shown in Fig. 6 should be included. The words "cell death was quantified by PI staining" is not informative. It is not known what kind of cell death was measured and how it was quantities.

4. Following above reasoning, data from H1299 KO cells reconstituted with a specific KEAP1 mutant found in lung cancer patients would be highly appreciated.

5. In Fig. 1, some details on cell viability measurement are to be added such as clonogenic survival and again measuring other kind of programmed cell death such as apoptosis and necroptosis would be highly appreciated based on the dynamic feature of lipid oxidative status.

6. Figs 1-2 show the cell ferroptosis cell death all in vitro cultured condition which could be more relevant if in vivo ferroptosis can be accessed since two uncontrolled conditions fatty acid oxidation and iron concentrations are undetermined in these in vitro culture conditions.

7. Fig. 2A shows Gene ontology analysis revealed ubiquinol synthesis pathway enrichment in KEAP1 mutant LUAD tumors. It would be strengthened if correlation of lack of ferroptosis pathways could be presented.

8. Fig. 3 shows only a related protein level, a direct KEAP1/ NRF2 complex formation and separation will be more informative.

9. Polyunsaturated fatty acids are susceptible to lipid peroxidation with their abundance and localization determining the amount of lipid peroxidation that occurs within the cell and therefore the extent of ferroptosis. It would be highly appreciated if any following up data is included with tumor response to RT or RT/immunotherapy.

10. Figure 4 shows that FSP1 is linked with KEAP1 deficient lung cancer but no direct link of KEAP1 and FSP1 is shown.

11. Fig. 6 showing tumor volumes of A549 xenografts that may add the data of KEAP1. It is strongly suggested that some non-ferroptosis cell death signature could be added to show the specificity of KEAP1 mediated radiosensitization.

Minor:

1. Abbreviations KEAP and NRF2 in Abstract may be defined.

2. Many descriptions are not precise or not using the scientific profession style, such as "KEAP1 is a tumor suppressor that normally acts as a substrate adaptor for the transcription factor NRF2 and targets it for proteasomal degradation by the KEAP1-Cullin-3 ubiquitin ligase complex under basal conditions". The grammar is not right and not clear what is the basal condition.

3. The Methods need to provide some details regarding the sample quality control.

4. Some terms such including CEL, RIN in Methods are to be defined.

Detailed Point-by-point response to the reviewer's comments:

Note to reviewers: We thank reviewers for taking efforts to review our manuscript and for providing insightful comments to further improve our manuscript. Below we provide the detailed point-by-point response to address all the comments raised by reviewers. To facilitate the review of our rebuttal letter and manuscript by reviewers, we present all the new data as rebuttal letter figures in this letter, with referrals to corresponding figures and text in our revised manuscript. We have also marked all the changes in our revised manuscript by colored text.

REVIEWER COMMENTS

Reviewer #1 (Remarks to the Author):

In the manuscript, Koppula and Lei et al. of the group of Boyi Gan present data on the CoQ-FSP1 axis that prevents ferroptosis and at the same time drives radiation resistance in lung cancer.

Ferroptosis is a recently discovered form of iron catalysed regulated necrosis (RN). In contrast to classical RN pathways such as necroptosis and pyroptosis, ferroptosis is not triggered by an active protein-mediated signalling but rather occurs if a failsafe, such as GPX4 or FSP1 cannot function properly. The two major anti-ferroptotic systems described so far include 1) the GPX4 system which requires glutathione (GSH) to prevent ferroptosis – it is inhibited by RSL3 and erastin. The second system, the FSP1 (AIFM2) oxidoreductase (as DHODH) reduce CoQ to CoQH2 which inhibits lipid peroxyl radical formation as an RTA. Importantly, this system prevents ferroptosis in a GSH-independent manner.

The tumour suppressor KEAP1, through the transcription factor NRF2, regulates expression of hundreds of proteins, many of which are involved in redox biology control. It has been known for some time that KEAP1 is inactive in some cancers, including lung cancers.

Here, the authors established a link between ferroptosis and KEAP1-deficient lung cancer and suggest that pharmacological inhibition of the CoQ-FSP1 axis sensitizes these cancers to radiation. This also works if the cancer cells are injected into mice (PDX). The authors conclude that this targeting might represent a novel therapeutic strategy to target KEAP1 mutant lung cancer.

Clearly, this topic is of interest to the readership of Nature Communications, and the advance in knowledge provided by this work is valuable. That said, I cannot recommend publication in a journal such as Nature Communications in the present form without some major concerns listed below being adequately addressed. For further improvement of the manuscript, I listed some minor remarks.

Thank you for having me as a referee for Nature Communications.

We thank this reviewer for the insightful comments and appreciate his/her comment that “this topic is of interest to the readership of Nature Communications, and the advance in knowledge provided by this work is valuable”. We hope that our revision now has fully addressed the critiques from this reviewer.

Major concerns

• The standard deviations in all experiments that demonstrate PI staining are surprisingly low. Are these really independent repetitions or are the same results simply measured several times? Of course, they should be repetitions performed with independently seeded cells. As an example for one figure, this applies to Figures 1b, 1c, 1d, 1k, 1l, 1p, but needs to be justified for all figures. A very good example of this concern is Figure 3m. Maybe, to be more convincing in this respect, authors might chose to demonstrate original FACS plots for some individual experiments.

We thank the reviewer for pointing this out, and would like to confirm that these data are PI staining analyses from independent repeats (rather than the same sample measured several times). Such small standard deviations have also been shown in other ferroptosis studies published in top journals (for example, see Fig. 1a, 1b, 1i, 2d, 3m in <https://www.nature.com/articles/s41586-019-1426-6>¹). To follow this reviewer's kind suggestion, we now show original FACS plots for Fig. 1C and Fig. 3L (Fig. 3M in the previous version) (**Rebuttal Figure 1; Rebuttal letter Fig. 1B** is shown as **Fig.S4H** in the manuscript. Due to space limit in the corresponding figure, we did not include rebuttal letter Fig. 1A in the manuscript, and only show it to the reviewer).

Figure 1. Original FACS plots from independent repeats for Fig. 1C (A) and Fig. 3L (B). Note the curve shape from FACS plots look differently among three repeats, showing that these data were generated from different samples (rather than the same sample measured three times).

• 4-Carboxybenzaldehyde (4CBA) is a derivate of benzaldehyde. Why would this by any means be specific for CoQ? Authos have to assess if 4CBA functions as a radical trapping agent which might explain the effects reported on the tumors in Fig. 6. Some cell lines do not express FSP1 and still do not die by ferroptosis spontaneously, but do so upon RSL3 or erastin treatment (Stockwell and Dixon data, Conrad data). Please also investigate the effects of 4CBA in settings that are independent of FSP1 to entirely rule out that 4CBA might inhibit GSH-dependent ferroptosis or ROS-induced cell death in general.

In our manuscript, we first introduced 4-CBA in the third paragraph of page 10: “KEAP1 deficiency-induced ferroptosis resistance to RSL3 or ML162 in these lung cancer cells was largely abolished under CoQ synthesis blockade conditions (by 4-chlorobenzoic acid [4-CBA] treatment).” As stated here, the full name of 4-CBA is 4-chlorobenzoic acid, not 4-Carboxybenzaldehyde. 4-CBA is a widely used inhibitor for CoQ biosynthesis enzyme COQ2,

and as far as we know, it has never been characterized as a radical trapping agent. We hope that this clarification clears the confusion from the reviewer on 4-CBA.

As for the second question from this reviewer “Please also investigate the effects of 4CBA in settings that are independent of FSP1 to entirely rule out that 4CBA might inhibit GSH-dependent ferroptosis or ROS-induced cell death in general”, in both FSP1 and our recent DHODH studies ²⁻⁴, 4-CBA has been used to show that inhibiting CoQ synthesis (by 4-CBA treatment) can largely abolish the ferroptosis sensitization effect caused by FSP1 or DHODH deletion, thereby making the point that the function of FSP1 (or DHODH) in regulating ferroptosis mainly depends on CoQ. Likewise, in our current study, we showed that the ferroptosis resistance phenotype caused by KEAP1 deficiency was largely abolished by 4-CBA treatment, suggesting that KEAP1 regulates ferroptosis mainly through CoQ. However, it is important to note that, while FSP1’s function in suppressing ferroptosis largely depends on CoQ, CoQ’s effect to suppress ferroptosis does not entirely depend on FSP1, because in the absence of FSP1, other enzymes (such as DHODH) can still reduce CoQ to CoQH₂ to defend against ferroptosis ⁴. Therefore, it is expected that 4-CBA should have FSP1-independent effects (in our study we propose that KEAP1-FSP1’s function to regulate ferroptosis depends on CoQ, but we did not claim that 4-CBA’s function in regulating ferroptosis depends on FSP1).

• *“Lipid peroxidation” is an interpretation (e.g. Fig. 5e, 5h, 5o and other). The axis should be labelled with what was actually measured.*

We thank the reviewer for the kind suggestion. As detailed in the method, we measured lipid peroxidation using C11-BODIPY staining followed by FACS analysis. In the ferroptosis field, it is accepted that the data from the C11-BODIPY staining can be presented as lipid peroxidation % or lipid ROS % (to give readers more straightforward information);

for example, see relevant publications from other top journals (**Rebuttal letter Fig. 2**, Panel A is from Fig. 2e in ¹; panel B is from Fig. 3d in ⁵).

• *It is currently under debate if ferroptosis is a pro- or an anti-inflammatory cell death (necroinflammatory potential of ferroptosis). Please discuss to which extent either hypothesis might influence the success of tumour rejection following radiation in your setting.*

This reviewer asked an interesting question. Ferroptosis has been considered as a form of pro-inflammatory cell death associated with the release of several immunostimulatory signals, such as damage-associated molecular pattern (DAMP) signals and lipid oxidation products, which can

Figure 2. Representative panels showing C11-BODIPY staining can be presented as lipid peroxidation % or lipid ROS %.

promote dendritic cell maturation and increase the efficiency of macrophages in the phagocytosis of ferroptotic cancer cells⁶. Notably, radiotherapy can also trigger pro-inflammatory processes through inducing immunogenic cell death, releasing immunostimulatory signals and reprogramming the permissive tumor microenvironment for immune-mediated tumor rejection⁷. Therefore, it is possible that the pro-inflammatory aspects of ferroptosis can further enhance radiotherapy-induced antitumor immunity, in line with the observation that cyst(e)inase (a ferroptosis inducer) boosted radiotherapy-mediated tumour rejection in immunocompetent mice⁸.

Minor remarks

- *Why is the normalized survival fraction (e.g. Fig. 5a) demonstrated with a log axis?*

A cell survival curve refers to a plot of the fraction of cells that survive in response to radiation (normalized by the fraction of cells that survive with no radiation exposure) versus the radiation dose. In the field of radiation biology, conventionally, the surviving fraction is always depicted on a logarithmic scale and plotted on the y-axis against doses on the x-axis. Based on the linear (the dose)-quadratic (logarithmic) model, radiation-mediated cell killing is not linear but exponential against different doses, so it has to be expressed in a logarithmic form in the y-axis⁹⁻¹². See the following link for detailed description https://en.wikipedia.org/wiki/Cell_survival_curve.

- *The list presented in Figure S1 is important and should and should not be hidden in the supplement of the manuscript.*

Taking the nice suggestion from this reviewer, we now moved Figure S1 into the main figure (Figure 1a).

- *The sentence “Accumulating evidence indicates that ferroptosis is a critical tumor suppression mechanism.” requires direct citations.*

This refers to the sentence at the beginning of page 4. We now added citations for this statement in our revised manuscript.

Reviewer #2 (Remarks to the Author):

KEAP1 mutations are frequently observed in NSCLC and lead to resistant to most standard cancer therapy. In this manuscript, Gan group found that KEAP1 deletion significantly promotes ferroptosis resistance to RSL3 or ML162, which is independent of SLC7A11 or GPX4. The FSP1–CoQ10–NAD(P)H pathway exists as a stand-alone parallel system and co-operates with GPX4 and glutathione to suppress ferroptosis. Here, Gan group found that FSP1 is a NRF2 transcriptional target and that KEAP1 deficiency in lung cancer cells leads to FSP1 upregulation through NRF2, resulting in ferroptosis resistance. They further demonstrated that the NRF2–FSP1 axis acts as a key downstream effector to mediate radioresistance in KEAP1 deficient lung cancer cells. Finally, in addition to in vitro cell culture experiments, via in vivo xenograft and PDX mouse models, they found that blocking FSP1–CoQ10–NAD(P)H axis by targeting CoQ synthesis (by 4-CBA treatment) can sensitize KEAP1 mutant or deficient tumors

to radiotherapy via inducing ferroptosis. This is well designed study with solid data to support the conclusion. Moreover, this study might provide another strategy to overcome radioresistance in KEAP1 mutant NSCLC, a deadly disease we are facing today. This manuscript is suitable for publication on Nature Communication.

We appreciate the positive and insightful comments from this reviewer. We hope that our revision now has addressed the critiques from this reviewer.

Since FSP1–CoQ10 axis regulates NAD(P)H production, which plays an important role against lipid peroxidation. It would be great if that the alteration of NADPH level can be examined when targeting FSP1 or CoQ1 in in vitro cell culture experiments.

This is a great suggestion! As shown in **Rebuttal letter Fig. 3** (Fig. S8 in the revised manuscript), we found that FSP1 deletion in A549 cells decreased NADP⁺/NADPH ratio, whereas FSP1 overexpression in H1299 cells had the opposite effect. These data are in line with the known role of FSP1 as an oxidoreductase that consumes NAD(P)H to reduce CoQ to CoQH₂^{2,3}.

In addition, it will interesting to know that whether 4-CBA or other compounds/chemicals that target FSP1–CoQ10–NAD(P)H axis to induce ferroptosis has been proposed or be in clinical trial for cancer treatment. This could be added in the discussion, which is essential to link current findings to translational cancer treatment.

We thank the reviewer for asking this interesting question. As far as we know, 4-CBA or iFSP1 (which targets the FSP1-CoQ10-NAD(P)H axis) has not been tested in any clinical trial yet. We hope that our preclinical studies presented in this study can motivate further research to test 4-CBA in clinical trial as well as the development of better compounds that target this signaling axis for clinical application in the future. We now added a brief discussion on this point (see the last paragraph in page 17).

Reviewer #3 (Remarks to the Author):

In this manuscript, the authors reported that FSP1 is an NRF2 target gene; and demonstrated that targeting FSP1 to include ferroptosis is a great strategy for treating KEAP1 mutant lung cancer. In particular, they found that inhibition of KEAP1 expression render lung cancer cells more resistance to class 2 FINs, but not to class 3 FIN, from which they identified two parallel pathways controlled by NRF2: (1) NRF2-SLC7A11-GPX4, and (2) NRF2-FSP-CoQ. Certainly this is a very significant and timely finding. Using H1299 xenograft model, the authors clearly demonstrated that knockout of FSP slowed down the growth of KEAP1-deficient tumors. Furthermore, using different lung cancer cell lines, the author demonstrated pharmacological and genetic deletion of FSP1 sensitizing cancer cells to radiation therapy (RT) in an NRF2-dependent manner. The translational value of this study was further demonstrated by the

Figure 3. The effects of FSP1 deletion in A549 (A) or FSP1 overexpression in H1299 (B) on NADP⁺/NADPH ratio. sg: sgRNA; C: control; ***, P<0.001; ****, P<0.0001.

synergistic effects of using CBA (an inhibitor of COQ2 that can be used in vivo) and RT as a combination therapy. In addition, CBA radiosensitized lung cancer cells but not normal lung epithelial cells (HBECS). Finally a lung cancer PDX model with KEAP1 mutation was further used to demonstrate the effectiveness of RT and 4-CBA combination therapy. The quality of the data are high with appropriate statistics. The topic is well suited to broad readership of this journal.

The following minor points should be addressed.

We appreciate the positive and insightful comments from this reviewer. We hope that our revision now has addressed the critiques from this reviewer.

*“sensitivity analyses of KEAP1 deficient lung cancer cells to different ferroptosis inducers link KEAP1 function in ferroptosis regulation to ubiquinone (CoQ) pathways”
The authors used both “KEAP1 deficient lung cancer cells” or “KEAP1 inactivated lung cancer cells” throughout the entire manuscript. The H1299 and H23 cell lines used in this study have no mutation in KEAP1 or NRF2 based on this paper. However the authors deleted KEAP1 using sgRNA. So in that sense, “KEAP1 deficient” should be used. However, in real world, we large see Keap1 mutated cancer cells, which would call for the use of “KEAP1 inactivated lung cancer cells”. So it is a tough call, how to label KEAP1-mutated cancer cells.*

We thank the reviewer for raising this point. In Results we used the term “KEAP1 deficiency/KO/deletion” when we refer to KEAP1 KO cells generated by CRISPR-Cas9 approach. In this study, we also used KEAP1 mutant cancer cells or tumors, which still express mutant (but inactive) KEAP1. Therefore, we used the term “KEAP1 inactive/inactivated lung cancers” to capture all these scenarios in the title/abstract/introduction of our manuscript.

*“the CoQ-FSP1 axis mediates ferroptosis- and radiation- resistance in KEAP1 deficient lung cancer cells”
The statement is not precise since the CoQ-FSP1 axis mediates ferroptosis- and radiation-resistance in any cells regardless the KEAP1 status. It is just the higher expression of FSP in cancer cells than normal cells make cancer cells more susceptible to FSP1 inhibition than normal cells.*

Here ferroptosis- and radiation-resistance refer to the more pronounced ferroptosis- and radiation-resistant phenotypes in KEAP1 deficient lung cancer cells compared to KEAP1 wild-type (WT) lung cancer cells. Our point is that these resistant phenotypes in KEAP1 deficient cells (relative to WT cells) are mediated by the CoQ-FSP1 signaling axis (because (1) FSP1 is upregulated in KEAP1 deficient cells, and (2) blocking FSP1 function can re-sensitize KEAP1 KO cells to ferroptosis and radiation). It is possible that inhibiting FSP1 function in KEAP1 WT cells (which exhibit low FSP1 expression) anyway would enhance ferroptosis- and radiation-sensitization. This remains to be tested.

From a therapeutic standpoint, since the CoQ-FSP1 axis mediates ferroptosis- and radiation-resistance in KEAP1 deficient lung cancer cells (which are generally resistant to radiotherapy), we propose to target this pathway as a therapeutic strategy to sensitize KEAP1 deficient tumors

to radiotherapy. On the other hand, if tumors (such as KEAP1 WT tumors) are already sensitive to radiotherapy, in clinic there is no point to add additional radiosensitizers to make such tumors even more sensitive to radiation.

*“Oxidative stress or KEAP1 inactivation (by its loss-of-function mutation, deletion, or epigenetic silencing in cancers) displaces NRF2 from KEAP1 and stabilizes NRF2”
The two-side binding model of NRF2-KEAP1 does not support “displaces NRF2 from Keap1” model, since the strong binding site ETGE-Kelch is hard to be displaced.*

We thank the reviewer for pointing this out. We have deleted “displaces NRF2 from Keap1” in this sentence and changed the statement to “Oxidative stress or KEAP1 inactivation (by its loss-of-function mutation, deletion, or epigenetic silencing in cancers) stabilizes NRF2”.

*“KEAP1 deletion in H1299 cells dramatically decreased GPX4 levels (Fig. 1A) in an NRF2-dependent manner, as deleting NRF2 in KEAP1 deficient H1299 cells restored GPX4 expression (Fig. S2B).”
It is interesting that NRF2 negatively regulating GPX4 levels in H1299, which is contradictory to the majority of papers published in the literature, reporting a positive correlation. This negative regulation of GPX4 by NRF2 should be discussed. Furthermore, it is intriguing to see that KEAP1 deletion increased GPX4 levels in H23 cells, whereas decreased GPX4 levels in H1299, however, their responses to all classes of FINs tested are very similar. Please explain?*

We thank the reviewer for asking this interesting question. In contrast to the expected results that KEAP1 deficiency should increase GPX4 levels, we found that in H1299 cells KEAP1 deletion actually decreased GPX4 expression in an NRF2-dependent manner. Consistent with this, another recent study showed that NRF2 knockdown increased GPX4 levels in some lung cancer cell lines (see Fig. 6A-B in

<https://www.sciencedirect.com/science/article/pii/S1097276520306936?via%3Dihub>)¹³.

However, in other lung cancer cell lines (such as H23 cells) we observed opposite results that KEAP1 deletion increased GPX4 levels. The mechanisms underlying these discrepancies in different lung cancer cell lines remain unknown but could relate to other genetic factors that might have influenced NRF2 regulation of GPX4. We now briefly discussed this interesting point in our revised manuscript (see the last paragraph in page 16).

Regardless the direction of GPX4 level change in these KEAP1-deficient lung cancer cells, KEAP1 deletion always leads to ferroptosis resistance to class 2 FINs (GPX4 inhibitors) but does not affect cellular sensitivity to the class 3 FIN (FIN56). Our interpretation is that KEAP1 deletion leads to upregulation of FSP1, which acts independent of GPX4 to suppress ferroptosis and therefore renders KEAP1-deficient lung cancer cells resistant to GPX4 inhibition (regardless of GPX4 level changes). Because FSP1 depends on CoQ to suppress ferroptosis, the increased FSP1 expression caused by KEAP1 deletion does not influence cellular sensitivity to FIN56 (which induces ferroptosis partly by depleting CoQ). We hope this reviewer will agree with our interpretation.

*“KEAP1 mutant H460 cells”
Please provide more information about the NRF2 and KEAP1 mutation status of H460.*

H460 carries a missense mutation in KEAP1 (D236H) but does not carry a mutation in NRF2. As shown in our recent publication¹⁴, H460 cells exhibit drastically increased levels of NRF2 and NRF2 targets such as SLC7A11 compared to other KEAP1 WT cells, showing that KEAP1 D236H mutant is inactive in H460 cells.

“Together, these results strongly suggest that KEAP1 deficiency upregulates FSP1 levels and that the CoQ-FSP1 signaling axis plays an important role in mediating ferroptosis resistance to class I FINs in KEAP1 deficient lung cancer cells.”
I think “class II FINs” should be used here.

We thank the reviewer for pointing this out. This statement indeed refers to Class 2 FINs (GPX4 inhibitors). We have corrected this in the manuscript.

Although it would not affect the conclusions of this manuscript, but using apigenin as an NRF2 inducer is not a good idea since it affects so many pathways. Left panel of S Figure 3F should be either deleted or the experiment is repeated using sulforaphane as an NRF2 inducer.

Taking the kind suggestion from this reviewer, we deleted Apigenin data from the manuscript.

“It should be noted that RT and 4-CBA treatment alone or in combination did not affect animal weight (Fig. S8E, F)”
I think the Y axis in S8F should be “Body weight (g)”.

We thank the reviewer for pointing this out. We have corrected this in the revised manuscript (now Fig. S7J in the manuscript).

Reviewer #4 (Remarks to the Author):

Radiotherapy (RT) is applied to treat more than half cancer patients worldwide; whereas the exact mechanism underlying radiation mediated tumor cell death, although extensively studied, remains unelucidated. One of the questions is whether ferroptosis, a specific programmed cell death due to imbalanced lipid oxidation and digestion. Using KEAP1 inactivated lung cancer cells, this study by Koppula et al, revealed that KEAP1, a stress responsive protein, can regulate ferroptosis by activating CoQ pathway. This is evidenced by inhibition of ferroptosis by FSP1 indicating the CoQ-FSP1 axis. KEAP1 is shown to regulate ferroptosis suppressor protein 1 (FSP1) via NRF2 transcription suggests is found to be CoQ-FSP1 axis that plays a mediates ferroptosis- and radiation- resistance in KEAP1 deficient lung cancer cells. Importantly, inhibition of the CoQ-FSP1 axis demonstrates a potential therapeutic effect of radiosensitization of KEAP1 inactivated lung cancer cells and PDX which are all impressive. However, some gaps in the description of experimental design and data explanation weakened the conclusion. The precise KEAP1 mediated NRF1 regulation is to be clarified.

We appreciate the positive and insightful comments from this reviewer. We hope that our revision now has fully addressed the critiques from this reviewer.

1. In Abstract, there are several gaps in describing the specific pathway. Why focusing on KEAP1 inactivated lung cancer is not justified. The ubiquinone (CoQ) is not logically linked to the study of NRF2. It is not clear how the CoQ-FSP1 axis is found. A cartoon illustrating the KEAP1 mediated ferroptosis may help to quickly see the innovative finding.

We thank the reviewer for the suggestion to improve the clarity of our manuscript. Taking this reviewer's kind suggestion, we have further revised our abstract and added a schematic in the revised manuscript (as Fig. 7) to summarize our model.

2. In Introduction, lung cancer patients with KEAP1 mutations have shorter overall survival with poor prognosis, and KEAP1 mutant lung tumors are refractory to most available standard-of-care therapies including RT, highlighting the pressing need to develop novel and effective combination therapies for this type of lung cancer. It is not clear what kinds of KEAP1 mutations are carried in these patients and whether the mutant forms are functional deficient or functional gained which is a fundamental issue for proposing the current study. Metadata analysis of lung cancer patient's outcome with different kind of KEAP1 mutants are to be illustrated.

KEAP1 is a well-established tumor suppressor in lung cancer. Most KEAP1 mutations in lung cancer (as well as in other cancers) are inactivating mutations, including truncating mutations and missense mutations that disrupt its normal functions (such as binding to NRF2). As far as we know, gain-of-function mutations for KEAP1 are rare in human cancers. Since our current study does not focus on KEAP1 mutation and patient outcome analyses and because multiple other studies have already done such analyses, we cited relevant papers in **Introduction**¹⁵⁻¹⁷ and refer readers to the cited literature for detailed information.

3. Fig. 1 is to reveal KEAP1 specifically involved in ferroptosis, thus other kinds of cell death should be considered as controls. In Fig1 legend, sg1 sg2 etc shown in the panel are not explained. In addition, in this figure, KEAP1 is studied in only one H1299 KEAP1 KO cell. At least another lung cancer cell line as shown in Fig. 6 should be included. The words "cell death was quantified by PI staining" is not informative. It is not known what kind of cell death was measured and how it was quantities.

To address this reviewer's comment, we also deleted KEAP1 in another KEAP1 WT lung cancer cell line H23 cells. As shown in **Rebuttal letter Fig. 4** (Fig. S1L-R in the manuscript), our data showed that KEAP1 deletion rendered H23 cells resistant to ferroptosis induced by erastin, RSL3, but not FIN56 (panels A-D), and that SLC7A11 knockdown promoted RSL3- or ML162-induced ferroptosis in H23 cells but not in KEAP1 KO counterparts (panels E-G). These data are consistent with those made in H1299 cells and substantiate our conclusion that KEAP1 deletion promotes ferroptosis resistance to class 2 FINs (RSL3 and ML162), but not to the class 3 FIN (FIN56).

Figure 4. The effects of KEAP1 deletion or SLC7A11 knockdown on the indicated protein levels and ferroptotic cell death in H23 cells upon indicated treatment. ***, $P < 0.001$, ****, $P < 0.0001$.

As this reviewer kindly suggested, we now provided the definition for sg1 (single guide RNA 1) in corresponding figure legends. Regarding the reviewer's question on other forms of cell death, ferroptosis was initially discovered from the studies characterizing cell death induced by erastin and RSL3 (which were later categorized as class 1 and 2 ferroptosis inducers, respectively)^{18,19}; therefore, by definition, erastin- or RSL3-induced cell death is ferroptosis, which is well recognized in the field of ferroptosis. In any case, we further confirmed that in H1299 cells (the cell line extensively used in this study), erastin- or RSL3-induced cell death can be rescued by the ferroptosis inhibitors (ferrostatin-1), but not by other cell death inhibitors (the apoptosis inhibitor Z-VAD-FMK and the necroptosis inhibitor Necrostatin-1s), demonstrating that erastin or RSL3 induces ferroptosis, but not other forms of cell death, in H1299 cells (**Rebuttal letter Fig. 5; Fig. S1D-E** in the revised manuscript).

Figure 5. The effects of cell death inhibitors on cell death (A) and cell viability (B-C) in H1299 upon erastin or RSL3 treatment. ****, $P < 0.0001$.

4. Following above reasoning, data from H1299 KO cells reconstituted with a specific KEAP1 mutant found in lung cancer patients would be highly appreciated.

To address this reviewer's comment, we re-expressed patient-derived KEAP1 G333C mutant in KEAP1 KO H1299 cells. As shown in **Rebuttal letter Fig. 6** (Fig. S3C, D in the manuscript), restoration of KEAP1 G333C mutant in KEAP1 KO H1299 cells failed to exert any rescuing effect on FSP1 or NRF2 levels, or erastin- or RSL3-induced ferroptosis. These results provide additional evidence to support our conclusion.

Figure 6. The effects of restoration of patient-derived KEAP1 G333C mutant in KEAP1 KO H1299 cells on NRF2 or FSP1 expression (A) and erastin- or RSL3-induced ferroptotic cell death (B). *: P<0.05, **: P<0.01, ***: P<0.001, ****: P<0.0001; ns: not significant.

5. In Fig. 1, some details on cell viability measurement are to be added such as clonogenic survival and again measuring other kind of programmed cell death such as apoptosis and necroptosis would be highly appreciated based on the dynamic feature of lipid oxidative status.

Cell viability was measured using the CCK8 kit (Dojindo, Japan). Cells were seeded into 96 well plates. Following appropriate treatment, cell culture media was replaced with media containing 10% CCK8 reagent. The plate was incubated for 1 hr at 37 °C in the CO₂ incubator and OD value at 450 nm was measured by microplate reader (FLUOstar Omega, BMG Labtech) to quantify cell viability. We now provided more detailed information in methods (see page 21-22 under “Cell death and viability assays”).

As discussed above, the cell death induced by these ferroptosis inducers is well recognized as ferroptosis in the field. We now further demonstrated this point in H1299 cells (by cell death inhibitor rescue experiments) by using both cell death and cell viability assays (**Rebuttal letter Fig. 5**; Fig. S1D-E in the revised manuscript).

6. Figs 1-2 show the cell ferroptosis cell death all in vitro cultured condition which could be more relevant if in vivo ferroptosis can be accessed since two uncontrolled conditions fatty acid oxidation and iron concentrations are undetermined in these in vitro culture conditions.

We thank the reviewer for asking this important question. In our study, we have conducted 4-hydroxy-2-noneal (4-HNE, a lipid peroxidation marker) IHC to characterize lipid peroxidation and ferroptosis in tumor tissues (see Fig. 4E-F, Fig. 6L-M, 6P-Q). However, we acknowledge that other biological effects (such as apoptosis) might also play a role in tumor growth changes as observed in our xenograft models (also see question 11 below). Therefore, we have been careful in making our conclusions from these in vivo assays; for example, in FSP1 xenograft

experiments (Fig. 4), we stated that our results “suggest that FSP1 promotes *KEAP1* deficient lung tumor growth likely through suppressing lipid peroxidation and ferroptosis”.

7. Fig. 2A shows Gene ontology analysis revealed ubiquinol synthesis pathway enrichment in KEAP1 mutant LUAD tumors. It would be strengthened if correlation of lack of ferroptosis pathways could be presented.

In our GO analyses, ferroptosis pathways are indeed not enriched among the overexpressed genes in *KEAP1* mutant LUAD tumors.

8. Fig. 3 shows only a related protein level, a direct KEAP1/NRF2 complex formation and separation will be more informative.

The point of Fig. 3 is to establish the *KEAP1*-NRF2-FSP1 signaling axis, in which we showed that (1) FSP1 is an NRF2 transcriptional target, (2) in *KEAP1* KO lung cancer cells, increased NRF2 levels leads to enhanced FSP1 expression, and (3) this signaling axis governs ferroptosis sensitivity in lung cancer cells. The *KEAP1*-NRF2 interaction has been well characterized from many previous studies. In addition, most of the experiments in this figure regards comparing phenotypes between *KEAP1* WT and KO cells; however, we will not be able to measure *KEAP1*-NRF2 interaction in *KEAP1* KO cells. With this clarification, we hope the reviewer will agree that it is unnecessary to measure *KEAP1*-NRF2 interaction in this figure.

9. Polyunsaturated fatty acids are susceptible to lipid peroxidation with their abundance and localization determining the amount of lipid peroxidation that occurs within the cell and therefore the extent of ferroptosis. It would be highly appreciated if any following up data is included with tumor response to RT or RT/immunotherapy.

In Fig. 6J-Q, using both cell line- and patient-derived xenograft models, we showed the efficacy of 4-CBA as a radiosensitizer in *KEAP1* mutant lung tumors, which correlated with lipid peroxidation levels (by 4-HNE staining), but not DNA damage levels (by phospho-H2AX staining) in these tumor samples, suggesting that 4-CBA sensitizes tumors to radiation potentially through inducing ferroptosis. We now further conducted cleaved caspase-3 staining in these tumor samples and showed that the radiosensitizing effect of 4CBA does not correlate with apoptosis induction in these tumor samples (see question 11 and **Rebuttal letter Fig. 7** below). It will be interesting to further test the combinations of 4-CBA and radiotherapy + immunotherapy in future preclinical investigations. We now briefly discussed this point in Discussion (see the second paragraph, page 17).

10. Figure 4 shows that FSP1 is linked with KEAP1 deficient lung cancer but no direct link of KEAP1 and FSP1 is shown.

The mechanistic link between *KEAP1* and FSP1 was established in Figure 3, which revealed a *KEAP1*-NRF2-FSP1 signaling axis to regulate ferroptosis: FSP1 is a NRF2 transcriptional target and *KEAP1* deficiency in lung cancer cells leads to NRF2 stabilization and FSP1 upregulation through NRF2-mediated transcription. FSP1 then promotes ferroptosis resistance through

reducing CoQ to CoQH₂. To guide readers better on our findings, we now also added a schematic (Fig. 7) in the revised manuscript to summarize our findings.

11. Fig. 6 showing tumor volumes of A549 xenografts that may add the data of KEAP1. It is strongly suggested that some non-ferroptosis cell death signature could be added to show the specificity of KEAP1 mediated radiosensitization.

Fig. 6 shows that inhibition of CoQ synthesis by 4-CBA treatment or COQ2 deletion sensitized KEAP1 KO or KEAP1 mutant cancer cells to radiotherapy through promoting ferroptosis. To follow the kind suggestion from this reviewer, we conducted cleaved caspase-3 IHC (to measure apoptosis) on tumor samples (from both A549 xenografts and PDXs) treated with vehicle, 4-CBA, radiotherapy, and 4-CBA + radiotherapy. As shown in **Rebuttal letter Fig. 7** (Fig. S7C, D, G, H in the revised manuscript), our results showed that 4-CBA treatment did not increase cleaved caspase-3 levels under both basal and radiotherapy conditions, suggesting that apoptosis does not mediate 4-CBA-induced radiosensitization (as expected, radiotherapy increased cleaved caspase-3 staining).

Figure 7. Representative images and quantification of cleaved caspase-3 staining in A549 xenografts (A-B) and PDX TC494 (C-D) with indicated treatment. **: P<0.01; ***: P<0.001; ns: not significant.

Minor:

1. Abbreviations KEAP and NRF2 in Abstract may be defined.

We now provided the full names of KEAP1 and NRF2 in the abstract.

2. Many descriptions are not precise or not using the scientific profession style, such as “KEAP1 is a tumor suppressor that normally acts as a substrate adaptor for the transcription factor NRF2 and targets it for proteasomal degradation by the KEAP1-Cullin-3 ubiquitin ligase complex under basal conditions”. The grammar is not right and not clear what is the basal condition.

We changed this sentence to “Tumor suppressor kelch-like ECH associated protein 1 (KEAP1) is a substrate adaptor in the KEAP1-Cullin-3 ubiquitin ligase complex, which targets nuclear factor erythroid 2-related factor 2 (NRF2) for proteasomal degradation under unstressed conditions” (page 4 in the manuscript).

3. The Methods need to provide some details regarding the sample quality control.

We described sample quality control in the “Quantification and Statistical analysis” section (page 28 in the manuscript). We have further revised the description and provided additional details.

4. Some terms such including CEL, RIN in Methods are to be defined.

We have carefully searched Methods but could not find these terms in it. If the reviewer can provide more detailed information (such as page and line information), we will be happy to further define them.

References:

- 1 Wu, J. *et al.* Intercellular interaction dictates cancer cell ferroptosis via NF2-YAP signalling. *Nature* **572**, 402-406, doi:10.1038/s41586-019-1426-6 (2019).
- 2 Bersuker, K. *et al.* The CoQ oxidoreductase FSP1 acts parallel to GPX4 to inhibit ferroptosis. *Nature* **575**, 688-692, doi:10.1038/s41586-019-1705-2 (2019).
- 3 Doll, S. *et al.* FSP1 is a glutathione-independent ferroptosis suppressor. *Nature* **575**, 693-698, doi:10.1038/s41586-019-1707-0 (2019).
- 4 Mao, C. *et al.* DHODH-mediated ferroptosis defence is a targetable vulnerability in cancer. *Nature* **593**, 586-590, doi:10.1038/s41586-021-03539-7 (2021).
- 5 Gascón, S. *et al.* Identification and successful negotiation of a metabolic checkpoint in direct neuronal reprogramming. *Cell stem cell* **18**, 396-409 (2016).
- 6 Xu, H., Ye, D., Ren, M., Zhang, H. & Bi, F. Ferroptosis in the tumor microenvironment: perspectives for immunotherapy. *Trends in molecular medicine* (2021).
- 7 Herrera, F. G., Bourhis, J. & Coukos, G. Radiotherapy combination opportunities leveraging immunity for the next oncology practice. *CA: a cancer journal for clinicians* **67**, 65-85 (2017).
- 8 Lang, X. *et al.* Radiotherapy and immunotherapy promote tumoral lipid oxidation and ferroptosis via synergistic repression of SLC7A11. *Cancer discovery* **9**, 1673-1685 (2019).
- 9 Zeman, E. M., Schreiber, E. C. & Tepper, J. E. in *Abeloff's Clinical Oncology* 431-460. e433 (Elsevier, 2020).
- 10 Marcus, K. J. & Haas-Kogan, D. Pediatric radiation oncology. *Oncology of Infancy and Childhood. Philadelphia, Saunders Elsevier*, 241-256 (2009).
- 11 Jäkel, O. Light-ion radiation therapy planning. (2014).
- 12 Bazan, J. G., Le, Q.-T. & Zips, D. in *IASLC Thoracic Oncology* 330-336. e332 (Elsevier, 2018).
- 13 Takahashi, N. *et al.* 3D Culture Models with CRISPR Screens Reveal Hyperactive NRF2 as a Prerequisite for Spheroid Formation via Regulation of Proliferation and Ferroptosis. *Mol Cell* **80**, 828-844 e826, doi:10.1016/j.molcel.2020.10.010 (2020).

- 14 Koppula, P. *et al.* KEAP1 deficiency drives glucose dependency and sensitizes lung cancer cells and tumors to GLUT inhibition. *iScience* **24**, 102649, doi:10.1016/j.isci.2021.102649 (2021).
- 15 Chen, X., Su, C., Ren, S., Zhou, C. & Jiang, T. Pan-cancer analysis of KEAP1 mutations as biomarkers for immunotherapy outcomes. *Annals of translational medicine* **8** (2020).
- 16 Zehir, A. *et al.* Mutational landscape of metastatic cancer revealed from prospective clinical sequencing of 10,000 patients. *Nature medicine* **23**, 703-713 (2017).
- 17 Ogawa, Y. & Atkinson, D. E. Interactions between citrate and nucleoside triphosphates in binding to phosphofructokinase. *Biochemistry* **24**, 954-958 (1985).
- 18 Dixon, S. J. *et al.* Ferroptosis: an iron-dependent form of nonapoptotic cell death. *Cell* **149**, 1060-1072, doi:10.1016/j.cell.2012.03.042 (2012).
- 19 Yang, W. S. *et al.* Regulation of ferroptotic cancer cell death by GPX4. *Cell* **156**, 317-331, doi:10.1016/j.cell.2013.12.010 (2014).

REVIEWERS' COMMENTS

Reviewer #1 (Remarks to the Author):

The manuscript has been successfully revised, and I recommend publication without much delay. It helps a lot to see the original FACS plots! However, I still did not find the clear experiment demonstrating that 4-CBA does NOT function as a radical trapping agent.

Reviewer #2 (Remarks to the Author):

The revised manuscript addressed all my previous comments. I don't have any other concerns for publication.

Reviewer #3 (Remarks to the Author):

The authors have addressed all my concerns and I recommend acceptance.

Reviewer #4 (Remarks to the Author):

My questions have been well addressed with new data and reasonable explanation. A clearly described signaling network illustrated in the added cartoon strengthens their conclusion. Thus, this work will have immediate impacts in the field of cancer radio-immunotherapy. The paper should be published.

Detailed Point-by-point response to the reviewer's comments:

Reviewer #1 (Remarks to the Author):

The manuscript has been successfully revised, and I recommend publication without much delay. It helps a lot to see the original FACS plots! However, I still did not find the clear experiment demonstrating that 4-CBA does NOT function as a radical trapping agent.

We thank the reviewer for the kind support. Regarding the experiment to test whether 4-CBA acts as a radical trapping antioxidant (or not), we apologize for not stating our reason why we did not perform this experiment more clearly. In ferroptosis research, if an agent is found to have an anti-ferroptosis effect, one would consider the possibility that that agent might act as a radical trapping antioxidant. For example, in a recent study (1), based on their observation that tetrahydrobiopterin (BH4) suppresses ferroptosis independent of its role as a cofactor, the authors tested and subsequently validated the hypothesis that BH4 acts as a radical trapping antioxidant. In our case, since 4-CBA (as a COQ2 inhibitor) has a pro-ferroptosis effect (that is, 4-CBA and radical trapping antioxidants have opposite effects on ferroptosis), there seems to be little rationale to test whether 4-CBA acts as a radical trapping antioxidant (or not). In our view, proving that 4-CBA does not function as a radical trapping agent would add little to its known effect to inhibit COQ2 and promote ferroptosis.

In addition, 2,2,-diphenyl-1-picrylhydrazine (DPPH) and fluorescence-enabled inhibited autoxidation assays are typically used to examine whether an agent has radical trapping antioxidant activities. If these are simple and well established assays in our lab, we would like to conduct these assays regardless of the rationale issue discussed above. However, these are not trivial assays, and only very selected experts in the field can conduct these analyses. It probably will take a few other months to establish these assays in our lab. Therefore, we kindly ask this reviewer to consider these two factors, and hope this reviewer will agree that, in the context of this study, it is not necessary to test whether 4-CBA acts as a radical trapping antioxidant or not.

Reviewer #2 (Remarks to the Author):

The revised manuscript addressed all my previous comments. I don't have any other concerns for publication.

We thank the reviewer for the kind support.

Reviewer #3 (Remarks to the Author):

The authors have addressed all my concerns and I recommend acceptance.

We thank the reviewer for the kind support.

Reviewer #4 (Remarks to the Author):

My questions have been well addressed with new data and reasonable explanation. A clearly

described signaling network illustrated in the added cartoon strengthens their conclusion. Thus, this work will have immediate impacts in the field of cancer radio-immunotherapy. The paper should be published.

We thank the reviewer for the kind support.

Reference:

1. Soula M, Weber RA, Zilka O, Alwaseem H, La K, Yen F, Molina H, Garcia-Bermudez J, Pratt DA, Birsoy K. Metabolic determinants of cancer cell sensitivity to canonical ferroptosis inducers. *Nature chemical biology*. 2020;16(12):1351-60 PubMed PMID: 32778843.